# DNA 5-methylcytosine detection and methylation phasing using PacBio circular consensus sequencing

Peng Ni [1,2,3,8], Fan Nie[1,2,3,8], Zeyu Zhong [1,3], Jinrui Xu[1,3], Neng Huang[1,3], Jun Zhang[1,3], Haochen Zhao[1,3], You Zou[1,3], Yuanfeng Huang[4], Jinchen Li [4,5], Chuan-Le Xiao [6] ✉, Feng Luo [7] ✉ & Jianxin Wang [1,2,3] ✉

Long single-molecular sequencing technologies, such as PacBio circular consensus sequencing (CCS) and nanopore sequencing, are advantageous in detecting DNA 5-methylcytosine in CpGs (5mCpGs), especially in repetitive genomic regions. However, existing methods for detecting 5mCpGs using PacBio CCS are less accurate and robust. Here, we present ccsmeth, a deep-learning method to detect DNA 5mCpGs using CCS reads. We sequence polymerase-chain-reaction treated and M.SssI-methyltransferase treated DNA of one human sample using PacBio CCS for training ccsmeth. Using long (≥10 Kb) CCS reads, ccsmeth achieves 0.90 accuracy and 0.97 Area Under the Curve on 5mCpG detection at single-molecule resolution. At the genome-wide site level, ccsmeth achieves >0.90 correlations with bisulfite sequencing and nanopore sequencing using only 10× reads. Furthermore, we develop a Nextflow pipeline, ccsmethphase, to detect haplotype-aware methylation using CCS reads, and then sequence a Chinese family trio to validate it. ccsmeth and ccsmethphase can be robust and accurate tools for detecting DNA 5-methylcytosines.

5-methylcytosine (5mC), the most common form of DNA methylation, is involved in regulating many biological processes[1]. In humans, most 5mCs occur at CpG sites, which are associated with embryonic development, diseases, and aging[2,3]. Bisulfite sequencing (BS-seq) is now the most widely used methodology for profiling 5mC methylation[4]. In a bisulfite-treated genomic DNA, unmethylated cytosines are converted to uracils, while methylated cytosines are unchanged[5]. Thus, the methylation status of a segment of DNA can be yielded at single-nucleotide resolution. However, bisulfite treatment damages the DNA, which further leads to DNA degradation and the loss of sequencing

diversity[6]. Recently, two bisulfite-free methods, ten-eleven translocation-assisted pyridine borane sequencing[7] (TAPS) and enzymatic methyl-seq[8] (EM-seq) were also developed, which are both reported to have more uniformly coverage and higher unique mapping rates than BS-seq. Like BS-seq, TAPS and EM-seq can be applied to both short-read sequencing and long-read sequencing[9–11]. However, all these methods need extra laboratory techniques, which further leads to extra sequencing costs.

Two major long-read sequencing technologies, PacBio single-molecule real-time (SMRT) sequencing and nanopore sequencing of

[1]School of Computer Science and Engineering, Central South University, Changsha 410083, China. [2]Xiangjiang Laboratory, Changsha 410205, China. [3]Hunan Provincial Key Lab on Bioinformatics, Central South University, Changsha 410083, China. [4]Bioinformatics Center, National Clinical Research Centre for Geriatric Disorders, Department of Geriatrics, Xiangya Hospital, Central South University, Changsha 410000, China. [5]Centre for Medical Genetics & Hunan Key Laboratory of Medical Genetics, School of Life Sciences, Central South University, Changsha 410000, China. [6]State Key Laboratory of Ophthalmology, Zhongshan Ophthalmic Center, Sun Yat-sen University, #7 Jinsui Road, Tianhe District, Guangzhou, China. [7]School of Computing, Clemson University, Clemson, SC 29634-0974, USA. [8]These authors contributed equally: Peng Ni, Fan Nie. ✉e-mail: xiaochuanle@126.com; luofeng@clemson.edu; jxwang@mail.csu.edu.cn

Oxford Nanopore Technologies (ONT), can directly sequence native DNA without PCR amplification[12,13]. DNA base modifications alter polymerase kinetics in SMRT sequencing and affect the electrical current signals near the modified bases in nanopore sequencing[13]. Thus, DNA base modifications can be directly detected from native DNA reads of SMRT and nanopore sequencing without extra laboratory techniques[12,13]. For nanopore sequencing, computational methods for 5mC detection either apply statistical tests to compare current signals of native DNA reads with an unmodified control (Tombo[14]), or use pre-trained Hidden Markov models (nanopolish[15]) and deep neural network models (Megalodon[16], DeepSignal[17]) without a control dataset. Previous studies have shown that methods using pre-trained models achieve high accuracies for DNA 5mC detection from human nanopore reads[18,19].

Pulse signals in SMRT sequencing, which are associated with the nucleotides in which the polymerization reaction is occurring[13,20], include the interpulse duration (IPD) and the pulse width (PW). IPD represents the time duration between two consecutive sequenced bases. PW represents the time duration of a base being sequenced[20]. Besides the sequenced nucleotides, base modifications would also influence pulse signals. Using the differences in pulse signals between modified and unmodified bases, methods for detecting 5mC and other base modifications from SMRT data have been developed[21]. However, due to the low signal-to-noise ratio, the reliable calling of 5mC using early version SMRT data requires high coverage of reads (up to 250×)[12,13]. Based on the fact that unmethylated CpGs in vertebrates often range over long hypomethylated regions, Suzuki et al. proposed AgIn, which improved the confidence of 5mCpG detection by combining the IPD features of neighboring CpGs from SMRT data[22]. Recently, the PacBio circular consensus sequencing (CCS) technique was presented[23], in which subreads generated from a circularized template in a single zero-mode waveguide (ZMW) are used to call a consensus sequence (CCS/HiFi read) with high accuracy. Using the new CCS technique, Tse et al. developed a convolutional neural network (CNN)-based method, called holistic kinetic model (HK model), for genome-wide 5mCpG detection in humans[24]. For a CCS read, the HK model first calculates the mean IPD and PW values of each base after aligning the subreads of the CCS read to the reference genome. Then, for each CpG site in the CCS read, the HK model organizes the mean IPD values, the mean PW values, and the sequence context surrounding the CpG into a feature matrix. At last, the HK model feeds the feature matrix into the CNN-based model to get a methylation probability of the CpG[24]. HK model achieves above 90% sensitivity and specificity on 5mCpG detection at read level (i.e., at single-molecule resolution). However, the HK model requires relatively high CCS subread depth (at least 20× passed subreads for one CCS) for accurate 5mCpG detection, which limits the insert size in library preparation, further limits the length of CCS reads. Following the HK model, PacBio proposed another CNN-based method primrose[25], which has been claimed to have 85% read-level accuracy on 5mCpG detection from long PacBio CCS reads. Moreover, PacBio provided primrose's companion script in pb-CpG-tools (https://github.com/PacificBiosciences/pb-CpG-tools) to predict the site-level methylation frequencies of CpGs. Similar to AgIn[22], for each CpG in the genome, pb-CpG-tools organize the read-level methylated probabilities of the CpG and its neighboring CpGs predicted by primrose as features. Then, pb-CpG-tools feed the features into a CNN-based model to get a predicted methylation frequency of the targeted CpG.

In this study, we propose ccsmeth, a deep-learning method to detect DNA 5mCpGs by using kinetics features (IPDs and PWs) of PacBio CCS reads. Using bidirectional Gated Recurrent Unit (GRU) and attention neural networks, ccsmeth detects methylation states of CpGs at both read level and genome-wide site level. To assess the performance of ccsmeth, we sequenced amplified and M.SssI-treated DNA of human sample NA12898 using PacBio CCS with 10 Kb insert size. We

also sequenced a human male sample SD0651_P1 using both PacBio CCS with 15 Kb insert size and BS-seq. Experiments on the sequenced datasets and publicly available datasets of HG002[23,26] and CHM13[27] show that ccsmeth achieves higher accuracies than the HK model and primrose for 5mCpG detection at read level. ccsmeth also achieves high correlations with BS-seq and nanopore sequencing at genome-wide site level. The results demonstrate that ccsmeth accurately detects methylation states of CpGs from long (≥10 Kb) CCS reads at both read level and site level.

Allele-specific methylation (ASM) occurs in both imprinting and non-imprinting regions, which are associated with complex diseases[28,29] and cancers[30]. Recent studies showed that both PacBio CCS sequencing and nanopore sequencing can be used for haplotype-aware genome assembly[31], variant calling[32,33], and methylation phasing[34–37]. Here, with the improved 5mCpG detection of ccsmeth, we further develop a Nextflow[38] pipeline called ccsmethphase to detect haplotype-aware methylation using CCS reads. We also sequence a Chinese family trio using PacBio CCS to validate ccsmethphase. Results on the tested datasets show that ccsmethphase can accurately detect genome-wide allele-specific methylation. Furthermore, we demonstrate that PacBio CCS is now a comprehensive and accurate technology for 5mCpG detection and methylation phasing even in repetitive genomic regions.

## Results
### The ccsmeth algorithm for 5mCpG detection
Recurrent neural network (RNN) and attention mechanism are widely used artificial neural networks in natural language processing[39,40]. Both RNN and attention mechanism have been applied in base modification detection from nanopore long reads[16,17,41]. Here, we propose ccsmeth, a deep-learning method that is composed of bidirectional GRU[42] and Bahdanau attention[43] networks, to detect CpG methylation from PacBio CCS reads. ccsmeth is designed to predict methylation states of CpGs at both read level and site level. For a targeted CpG, ccsmeth first predicts the methylation probability (or binary methylation state) of the CpG in a read (i.e., at single-molecule resolution, read level), and then summarizes the read-level methylation states to get its methylation frequency (site level) in the targeted genome (Fig. 1, Methods). During the generation of a CCS read, the IPD and PW values of each base in forward and reverse complement strands of the CCS read are averaged from corresponding subreads (Fig. 1a). To predict the read-level methylation state of a CpG, ccsmeth extracts a 21-mer sequence context that includes the CpG itself in the center, with the kinetics information (the averaged IPD, the averaged PW and the number of covered subreads) of each base. Since CpG methylation are mostly symmetric in human[44], ccsmeth constructs two feature matrixes from the forward and reverse complement strand for a symmetric CpG pair (Fig. 1b). After processing the feature matrixes, ccsmeth outputs a read-level methylation probability $P_r$ ($P_r \in [0,1]$).

Before calling methylation at the site level, the CCS reads should be aligned to the reference genome. In ccsmeth, we provide two modes to infer the site-level methylation frequency of CpGs: count mode and model mode (Methods, Supplementary Fig. 1). In count mode, based on read-level methylation probabilities, binary methylation state (0 as unmethylated, 1 as methylated) of a CpG in per read is set by a probability cutoff (0.5 as default). Then the methylation frequency is calculated by counting the number of reads where the CpG is predicted as methylated, and the total number of reads mapped to the CpG. In the model mode of ccsmeth, we leverage the read-level methylation probabilities of neighboring CpGs to increase the confidence of the site-level methylation detection in a way similar to pb-CpG-tools. Specifically, for a targeted CpG, the read-level methylation probabilities of the CpG and its 10 adjacent CpGs, together with the distance (in base pair) of all 11 CpGs to the targeted CpG are organized into a feature matrix. The feature matrix is first input into a BIGRU layer

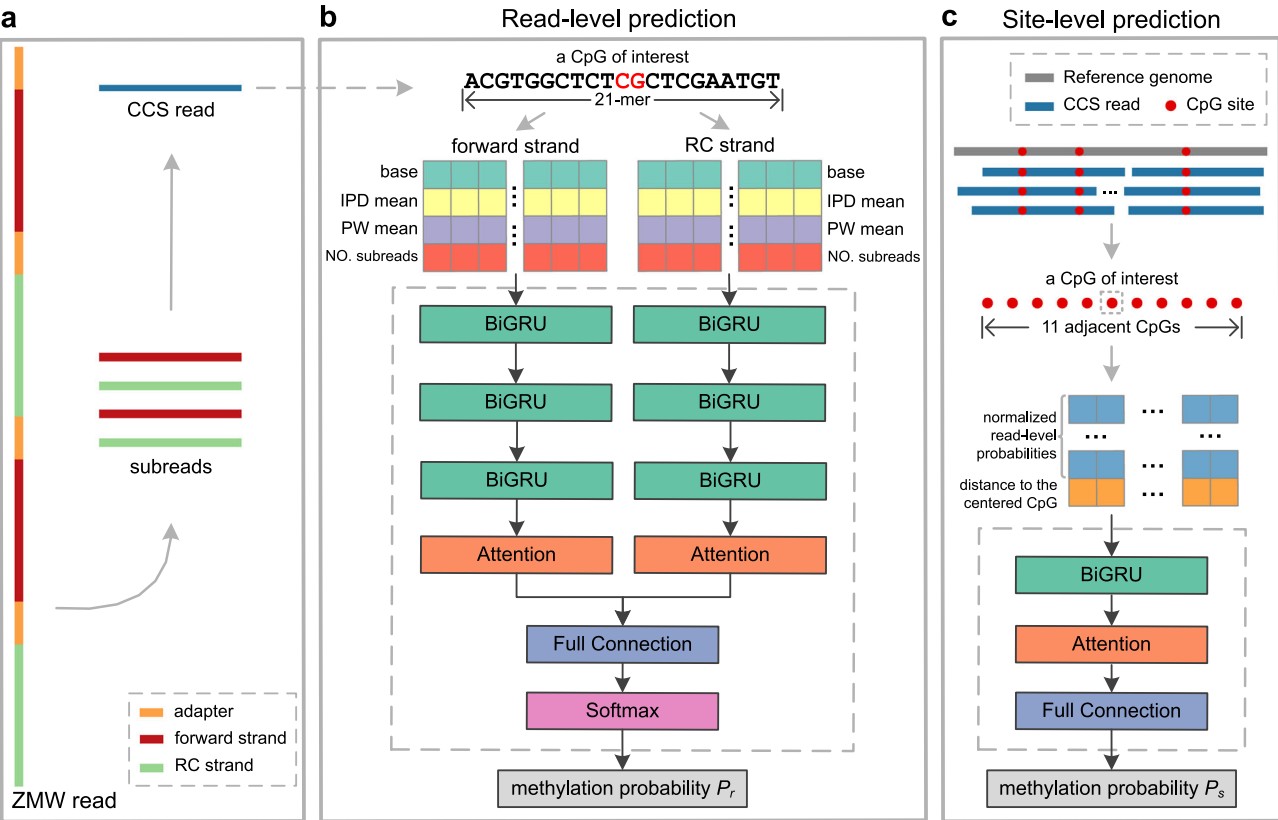

**Fig. 1 | ccsmeth for 5mCpG detection using PacBio CCS reads. a** Illustration of PacBio CCS. **b, c** Schema of ccsmeth to predict CpG methylation at read level and site level. RC reverse complement, BiGRU Bidirectional Gated Recurrent Unit layer, Full Connection fully connected layer, Softmax Softmax layer.

to capture the forward and reverse flow of interactions between adjacent CpGs. Then an attention layer is applied to optimize the weights of each adjacent CpGs, which allows the model to focus on the most relevant interactions. A methylation probability $P_s$ ($P_s \in [0, 1]$) is finally outputted as the methylation frequency of the targeted CpG (Fig. 1c, Supplementary Fig. 1).

## ccsmeth accurately detects CpG methylation at single-molecule resolution

To evaluate ccsmeth at read level (i.e., at single-molecule resolution), we first use three groups of CCS datasets (M01&W01, M02&W02, M03&W03) sequenced using M.SssI-treated and PCR-treated human DNA on different versions of PacBio sequencers[24]. In M.SssI-treated DNA, the CpG methyltransferase M.SssI methylates all CpGs, while PCR-treated DNA which is prepared via whole genome amplification (WGA) contains nearly no methylated bases[24]. As shown in Supplementary Table 1, M01-03 are M.SssI-treated DNA samples, and W01-03 are PCR-treated DNA samples. For each group of datasets, we randomly select 50% methylated reads and unmethylated reads for model training. The remaining 50% methylated and unmethylated reads are used for testing. We use the same reads to train and test the HK model[24]. primrose does not provide the interface for training, so we exclude it for comparison on these three datasets. As shown in Fig. 2a, ccsmeth outperforms the HK model on all three datasets. ccsmeth achieves accuracies of 0.9232, 0.8788, and 0.8765 on M01&W01, M02&W02, and M03&W03, respectively. The accuracies of ccsmeth are 5.4%, 4.4%, and 3.7% higher than HK model on the three datasets, respectively. ccsmeth achieves either around or above 0.95 AUCs, which are 3.3%, 3.4%, and 2.6% higher than HK model on the three datasets, respectively.

The read lengths of the three CCS datasets from Tse et al.[24] are all less than 10Kb, while CCS reads used in practice are usually in 10−25 Kb[23]. Therefore, we further use the long (≥10 Kb) CCS reads of three human samples (NA12898, HG002, and SD0651_P1) for read-level evaluation (Methods, Supplementary Table 2): The CCS reads of NA12898 are sequenced using PCR-treated and M.SssI-treated DNA of NA12898 with 10 Kb insert size; The CCS reads of HG002 native DNA sequenced using three different insert sizes (15Kb, 20Kb, 24Kb) were taken from Baid et al.[26] and Human Pangenome Reference Consortium[23]; The CCS reads of SD0651_P1 are sequenced using 15Kb DNA insert size. The mean subread depths of the CCS reads in these datasets range from 7.6× to 14.1× (Supplementary Table 1). We train the read-level model of ccsmeth using NA12898 CCS reads aligned to autosomes of the reference genome and one SMRT cell of HG002 CCS reads (Methods). The CCS reads of NA12898 aligned to chrX, 6 SMRT cells of HG002 CCS reads and the SD0651_P1 CCS reads are used for testing (Methods). We run primrose with its built-in model on the same testing data for comparison. As shown in Fig. 2b, ccsmeth gets 0.8721-0.9062 accuracies, 0.8621−0.8903 sensitivities, 0.8765−0.9220 specificities, and 0.9464-0.9682 AUCs, which are higher than those of primrose on all five datasets. Especially, ccsmeth gets much higher accuracies and specificities on HG002 CCS reads of 15 Kb, 20 Kb, and 24 Kb insert sizes: >4% higher accuracies and >7% higher specificities than those of primrose. To compare ccsmeth with the HK model, we subsampled 100 K ZMW reads from the datasets of NA12898 and three HG002 insert sizes (15 Kb, 20 Kb, 24 Kb) since HK model is extensively time-consuming on large datasets. The results show that HK model achieves similar accuracies with primrose, especially on the HG002 datasets. Accuracies of both HK model and primrose are lower than that of ccsmeth (Supplementary Fig. 2). Besides evaluating ccsmeth genome widely, we also evaluate ccsmeth in specific genomic contexts and regions to explore whether the performance of ccsmeth is correlated with any genomic features (Supplementary Note 1). As shown in Supplementary Fig. 3, ccsmeth

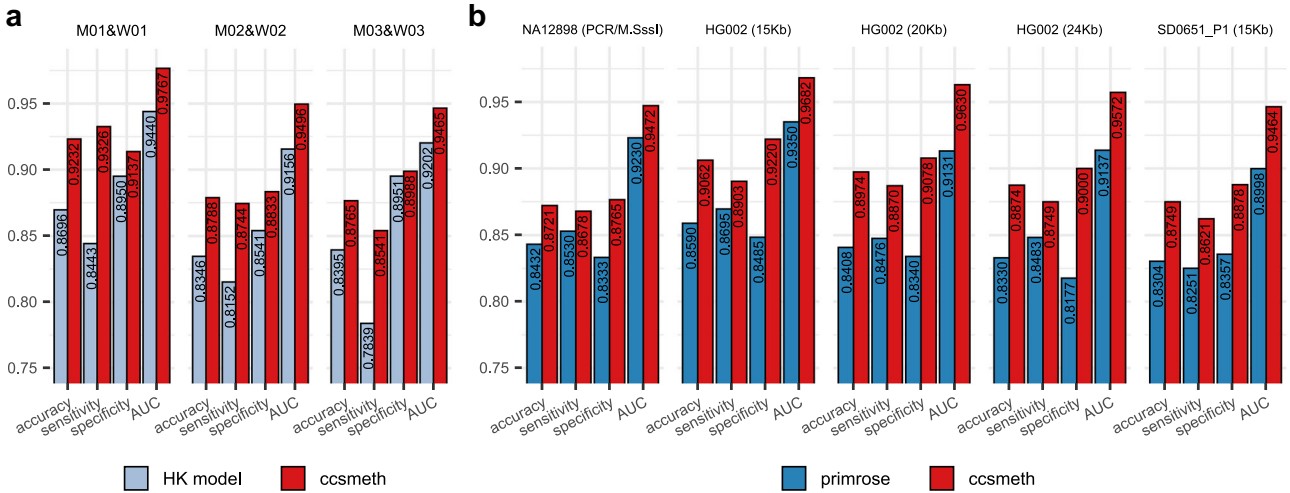

**Fig. 2 | Evaluation of ccsmeth on 5mCpG detection at read level. a** Comparing ccsmeth and HK model on three datasets of PCR-treated and M.SssI-treated human DNA. **b** Comparing ccsmeth and primrose on NA12898 (10 Kb, PCR/M.SssI-treated), HG002 (15 Kb, 20 Kb, 24 Kb), and SD0651_P1 (15 Kb) CCS reads. Values in the figure are the average of 5 repeated tests. AUC area under the curve. The standard deviation values of the multiple repeated tests are in Supplementary Table 4. Source data are provided as a Source Data file.

outperforms primrose in all tested regions. The results also show that ccsmeth tends to have higher accuracies in regions with high CpG densities, but has relative lower accuracies in intergenic regions, CpG shores, CpG shelves, and some repetitive regions.

The read-level accuracy of ccsmeth can be further improved by filtering out ambiguous calls. As shown in Supplementary Fig. 4a, by filtering out the calls with methylation probability close to 0.5, the incorrect calls of ccsmeth can be reduced. We define $\Delta_p = |P_r - P_r'|$ to filter out the ambiguous calls, where $P_r$ is the methylation probability, and $P_r'$ is the unmethylated probability defined as $1 - P_r$. Like modbam2bed[45], we set $\Delta_p$ to 0.33 for testing. Supplementary Fig. 4b shows that when $\Delta_p$ is set to 0.33, the accuracies of ccsmeth improve by 3.5–4.2% with 8.9–12.8% of calls being discarded.

## ccsmeth accurately detects CpG methylation at genome-wide site level

We use the CCS reads of HG002, SD0651_P1, and CHM13 to evaluate ccsmeth on 5mCpG detection at the site level (Methods, Supplementary Table 1 and 2). 2 SMRT cells of HG002 CCS reads are used to train the site-level model of ccsmeth; 6 SMRT cells of HG002 CCS reads (two for each of three insert sizes 15Kb, 20Kb, 24Kb), 2 SMRT cells of SD0651_P1 15Kb reads, and 2 SMRT cells of CHM13 20Kb reads are used for testing. There are 25.6×, 17.0×, 28.4×, 19.6×, and 16.5× average genome coverage of HG002 (15Kb, 20Kb, 24Kb), SD0651_P1 and CHM13 CCS reads used for testing in total, respectively. We downloaded BS-seq and nanopore R9.4.1 sequencing data of the three human samples as the benchmark (Supplementary Table 3). When evaluating ccsmeth, we subsample reads of the five datasets under certain coverages and compare the site-level results of ccsmeth and primrose with the results of BS-seq and nanopore sequencing (Fig. 3a–d). We repeat the subsampling of each coverage 5 times and get averaged values of metrics for comparison. The results show that the model mode of ccsmeth and primrose/pb-CpG-tools achieve higher Pearson correlations with BS-seq and nanopore sequencing than the count mode of ccsmeth and primrose/pb-CpG-tools do. Meanwhile, by setting $\Delta_p$ to 0.33 to filter out ambiguous calls, the Pearson correlations of the count mode of ccsmeth with BS-seq and nanopore sequencing improve by ~1-2% (Fig. 3a–d). On all tested datasets, ccsmeth achieves higher correlations than primrose/pb-CpG-tools does in both modes, especially under low coverages. For example, using 10× HG002 15 Kb, 20 Kb, and 24 Kb CCS reads, ccsmeth in

model mode obtains 0.9198, 0.9083, and 0.9087 correlations with BS-seq, while primrose/pb-CpG-tools in model mode only obtains 0.8864, 0.8653, and 0.8696 correlations, respectively. (Fig. 3a). ccsmeth in model mode also obtains 0.9062, 0.8967, and 0.8952 correlations with nanopore sequencing when using 10× HG002 15 Kb, 20 Kb, and 24 Kb CCS reads, which are 4.3%, 5.5%, and 4.9% higher than those obtained by primrose/pb-CpG-tools in model mode, respectively (Fig. 3b). On all tested datasets, ccsmeth also gets lower root mean square errors (RMSEs) than primrose/pb-CpG-tools gets in most cases (Supplementary Tables 5–12).

The model mode of ccsmeth can also be applied to the read-level results of primrose. As shown in Supplementary Tables 5–12, primrose with ccsmeth in model mode gets higher correlations and lower RMSEs with both BS-seq and nanopore sequencing than primrose with pb-CpG-tools in count mode gets. Especially, under low coverages (<15×) of HG002 and CHM13 datasets, primrose with ccsmeth in model mode outperforms primrose with pb-CpG-tools in model mode (Supplementary Tables 5–10 and 12). These results further demonstrate the effectiveness of the site-level model of ccsmeth.

We further test ccsmeth using the total CCS reads of HG002 15 Kb, 20 Kb, 24 Kb, SD0651_P1, and CHM13. Using the reads of HG002 15 Kb, 20 Kb, and 24 Kb datasets, ccsmeth gets 0.9463, 0.9271, and 0.9410 correlations with BS-seq, and gets 0.9287, 0.9127, and 0.9240 correlations with nanopore sequencing, respectively (Fig. 3e). When combining the total 71.0× CCS reads of HG002, ccsmeth achieves 0.9571 and 0.9394 correlations with BS-seq and nanopore sequencing, respectively (Supplementary Fig. 5, Supplementary Data 1). ccsmeth gets 0.8750 correlation with BS-seq using the total SD0651_P1 reads, and gets 0.9328 correlation with nanopore sequencing using total CHM13 reads (Fig. 3f, g, Supplementary Fig. 6). The results of ccsmeth on the three HG002 datasets are also highly correlated with each other (correlations>0.9344), which show the reproducibility of ccsmeth (Fig. 3e). We further use the 71.0× HG002 CCS reads to explore of which CpG contexts are predicted more accurately by the model mode in terms of methylation frequencies (Supplementary Note 2 and Supplementary Fig. 7). We classify the CpGs into two groups $G_m$ and $G_c$. $G_m$ contains CpGs whose methylation frequencies are more accurately predicted by model mode, while $G_c$ contains CpGs whose methylation frequencies are more accurately predicted by count mode. We find that the CpGs in $G_m$ tend to have either very low (<0.2) or high (>0.8) methylation frequencies.

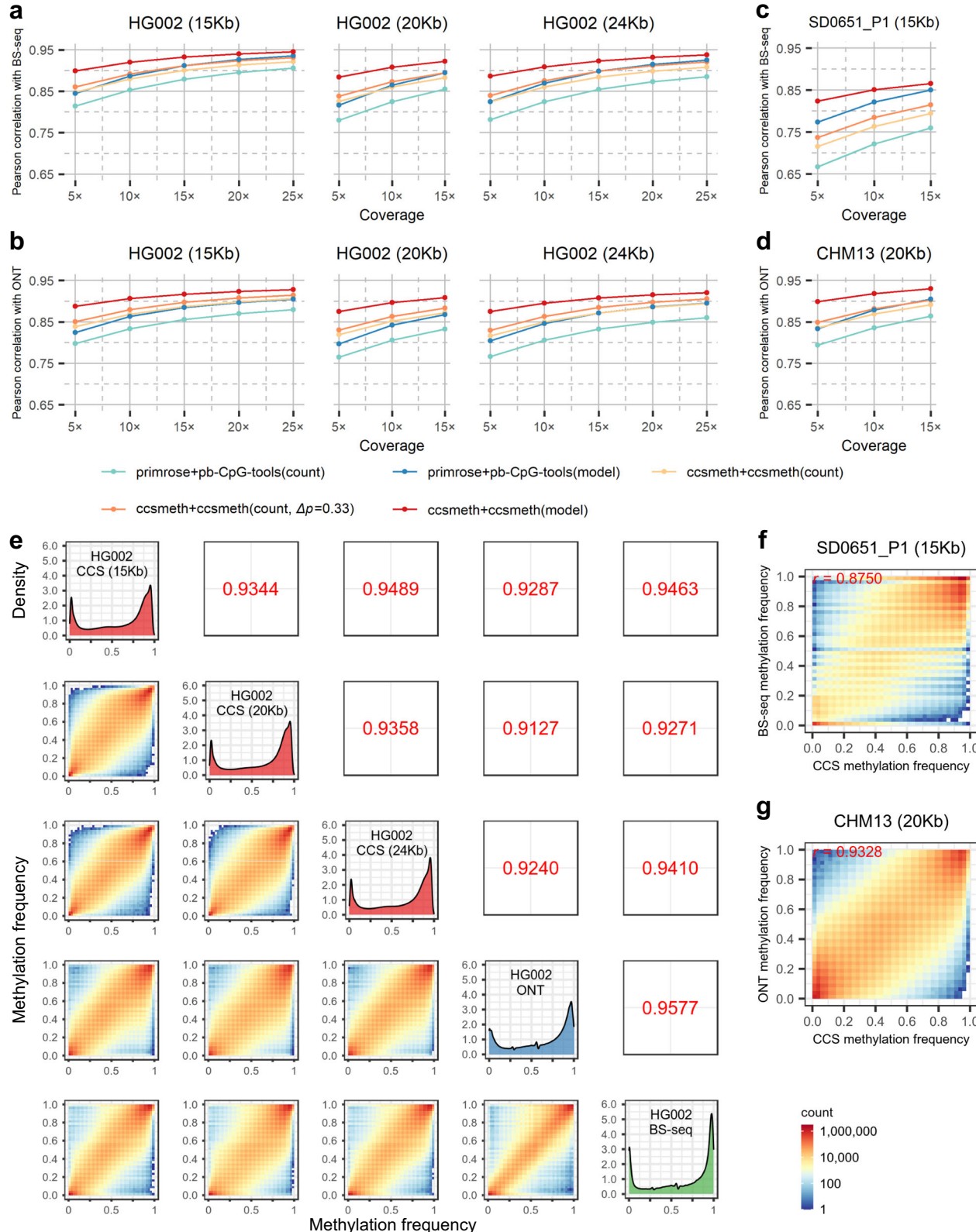

**Fig. 3 | Evaluation of ccsmeth on 5mCpG detection at genome-wide site level.** **a–d** Comparing ccsmeth and primrose/pb-CpG-tools against BS-seq and nanopore sequencing under different coverages of HG002, SD0651_P1, and CHM13 CCS reads. $\Delta_p$: Difference absolute value between methylated and unmethylated probabilities. Values are the average of 5 repeated tests. The standard deviation values of the multiple repeated tests are in Supplementary Tables 5–12. **e** Evaluation of ccsmeth model mode against BS-seq and nanopore sequencing using total CCS reads of HG002 (15Kb) (25.6×), HG002 (20 Kb) (17.0×), and HG002 (24 Kb) (28.4×), respectively. Values in upper triangles are Pearson correlations. CCS PacBio CCS sequencing; ONT nanopore sequencing, BS-seq bisulfite sequencing. **f** Evaluation of ccsmeth model mode against BS-seq using total 19.6× SD0651_P1 (15 Kb) CCS reads. *r*: Pearson correlation. **g** Evaluation of ccsmeth model mode against nanopore sequencing using total 16.5× CHM13 (20 Kb) CCS reads. Source data underlying **a**, **b**, **c**, and **d** are provided as a Source Data file.

## Haplotype-aware methylation calling and ASM detection using PacBio CCS data

Following ccsmeth, we further develop a Nextflow[38] pipeline called ccsmethphase for haplotype-aware methylation calling and ASM detection using only PacBio CCS data (Fig. 4a, Methods). In this pipeline, ccsmeth is used to call methylation states. Clair3[33] is used to call single nucleotide variants (SNVs). The SNVs called by Clair3 are then phased by WhatsHap[46] to generate haplotypes. DSS[47] is used to detect differentially methylated regions (DMRs) between two haplotypes.

We evaluate ccsmethphase with the total 71.0× HG002 CCS reads (Supplementary Table 2). We also use the HG002 BS-seq data (Supplementary Fig. 8, Supplementary Note 3) and nanopore data (Supplementary Fig. 9, Supplementary Note 4) to phase CpG methylations for comparison. First, we investigate the haplotype-aware methylation status of known imprinted regions using PacBio CCS data. We get 204 known imprinted intervals from Akbari et al.[48], in which there are 102 well-characterized imprinted intervals[49–53] (Methods). We compare the methylation difference of each imprinted interval between the two haplotypes of HG002 (Methods). As shown in Fig. 4b, the well-characterized imprinted intervals have large methylation differences between the two haplotypes (median = 0.53), while 22.1% of other known imprinted intervals also show large (>0.5) methylation differences. The methylation differences of known imprinted intervals got from CCS data are highly consistent with those got from BS-seq and nanopore data: Pearson correlations are 0.8605 and 0.9806, respectively (Supplementary Fig. 10 and 11). We examine the known imprinted intervals on SD0651_P1 CCS data and get consistent results (Supplementary Fig. 12a).

We then assess ccsmethphase on ASM detection using the HG002 sequencing data. Using the CCS reads, ccsmethphase generates 14,390 DMRs. Using the BS-seq and nanopore reads with corresponding pipelines, 2463 and 16,250 DMRs are generated, respectively. 81.4% DMRs generated using BS-seq reads are closely next to the genomic locations of the CCS-generated DMRs (distance<10 kb), and 70.8% of the DMRs overlap with the CCS-generated DMRs (Fig. 4c). Among the DMRs generated using nanopore reads, 68.8% DMRs are closely next to the genomic locations of the CCS-generated DMRs, and 51.7% DMRs overlap with the CCS-generated DMRs (Fig. 4d). Most of the CCS-generated DMRs are also closely next to the genomic locations of the DMRs generated using BS-seq and nanopore data in HG002 (Supplementary Fig. 13). From the SD0651_P1 CCS reads, ccsmethphase generates 8,183 DMRs. In both HG002 and SD0651, most of the known imprinted intervals are either overlapped with or near the CCS-generated DMRs (Fig. 4e, Supplementary Fig. 12b and 14), which also shows the ability of ccsmethphase on ASM detection. We also assess ccsmethphase on ASM detection using the CCS data of a Chinese family trio, in which HN0641_FA is the father, HN0641_MO is the mother, and HN0641_S1 is the son. The results show that ccsmethphase not only can detect known imprinted intervals but also reveals the patterns of parental imprinting correctly (Supplementary Note 6, Supplementary Figs. 15–18).

We further compare genome-wide site-level methylation frequencies of CpGs at two haplotypes detected by PacBio CCS data with those by BS-seq and nanopore data. To accomplish this, we would expect consistent haplotype assignment (*i.e.*, all maternal SNVs are assigned to one haplotype, and all paternal SNVs are assigned to another haplotype). However, because of the uneven reads coverage across the reference genome, we can only generate discrete haplotype blocks when using reads of a single sample to phase SNVs[34]. Therefore, we use the phased SNVs generated by Illumina trio data of HG002 to phase the CCS and nanopore reads. We compare the methylation frequencies of the phased CpGs predicted using CCS reads with those using BS-seq and nanopore reads. PacBio CCS gets >0.93 correlations with BS-seq and nanopore sequencing in both maternal and paternal haplotypes (Fig. 4f, g, Supplementary Table 13). This result further demonstrates that ccsmethphase can accurately detect haplotype-aware methylation in the human genome using CCS reads.

## Assessment of PacBio CCS for methylation detection and phasing in repetitive genomic regions

With longer reads, PacBio CCS is expected to profile methylation of more CpGs in the human genome than short-read sequencing technologies do. Using T2T-CHM13[27] (T2T: Telomere-to-Telomere) as the reference genome, we first assess the number of CpGs covered by HG002 CCS reads, especially in repetitive genomic regions: repetitive genomic elements annotated by RepeatMasker[54,55], segmental duplications (SDs)[56], and peri/centromeric satellites (cenSats)[57]. We also assess the total HG002 BS-seq and nanopore reads for comparison (Supplementary Table 3). As shown in Fig. 5a, using 15× coverage of CCS reads, 32.85 M (96.9%) of human CpGs are covered, of which there are 31.33 M CpGs covered by at least 5 reads. The CpGs covered by 15× CCS reads are more than the CpGs covered by 117.5× Illumina BS-seq reads. When using all 71.0× testing CCS reads, 32.74 M (96.6%) CpGs in the human genome are covered by at least 5 mapped reads, which are almost the same as the number of CpGs covered by 65.8× nanopore reads (Fig. 5a). In RepeatMasker repeats, SDs, and cenSats, PacBio CCS detects methylation states of 96.8%, 88.4%, and 85.3% CpGs, respectively. Compared to BS-seq, methylation states of 10.4%, 33.6%, and 34.7% CpGs in RepeatMasker repeats, SDs, and cenSats can only be detected by using CCS, respectively (Fig. 5b). In non-RepeatMasker regions, PacBio CCS detects methylation states of 96.1% CpGs, which are 8.9% more than the CpGs detected by BS-seq (Supplementary Fig. 19a).

The HG002 CCS reads are shorter than the HG002 nanopore reads (mean read length: 18,797 bp vs. 21,933 bp). However, the number of CpGs phased by PacBio CCS (*i.e.*, CpGs covered by at least 5 phased CCS reads) is not significantly less than the number of CpGs phased by nanopore sequencing: 26.97 M vs. 27.66 M (Fig. 5c). Both PacBio CCS and nanopore sequencing phase much more CpGs than BS-seq. with limited read length, BS-seq can only phase 6.71 M of human CpGs. PacBio CCS phases 85.4%, 60.2%, and 46.5% of the CpGs in RepeatMasker repeats, SDs, and cenSats, which are 63.8%, 45.9%, and 35.7% more than the CpGs phased by BS-seq, respectively (Fig. 5d). In non-RepeatMasker regions, PacBio CCS phases 64.4% more CpGs than BS-seq does (Supplementary Fig. 19b). Notably, PacBio CCS phases slightly more CpGs than nanopore sequencing in cenSats, which may indicate that highly accurate CCS reads are more suitable for SNV detection and methylation phasing across peri/centromeric regions than nanopore R9.4.1 reads are.

The methylation frequencies of CpGs predicted using PacBio CCS in repetitive genomic regions are highly correlated with those predicted using BS-seq and nanopore sequencing. In the RepeatMasker repeats, SDs, and cenSats of HG002, PacBio CCS gets 0.9540, 0.9208, and 0.8822 correlations with BS-seq, and gets 0.9358, 0.9087, and 0.8572 correlations with nanopore sequencing, respectively (Supplementary Table 16). For the haplotype-aware methylation detection in the repetitive genomic regions of HG002, PacBio CCS also gets >0.89 and >0.90 correlations with BS-seq and nanopore sequencing, respectively (Supplementary Table 17). In summary, with ccsmeth and ccsmethphase, PacBio CCS can be a comprehensive and accurate technology for 5mCpG detection and methylation phasing in repetitive genomic regions.

## Discussion

Due to its highly accurate long reads, PacBio CCS is becoming more widely used in genomics research, such as genome assembly[31], SNV detection[33], and structural variant (SV) detection[35]. However, compared to nanopore sequencing, the application of PacBio CCS on DNA 5mC detection and methylation phasing has not been fully studied before. In this study, we have developed and validated ccsmeth, a deep-learning method to detect 5mCpGs from PacBio CCS reads.

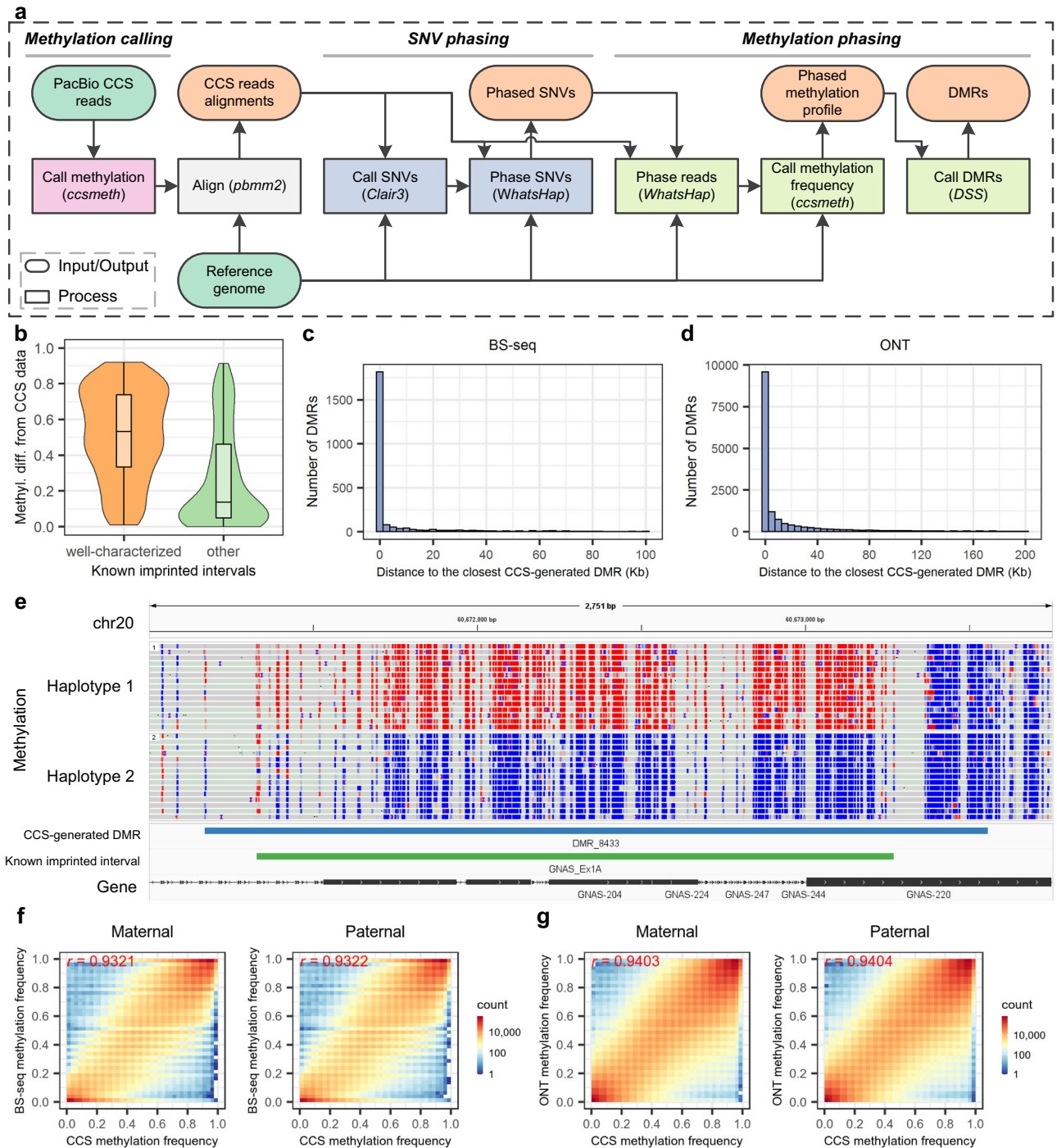

**Fig. 4 | Methylation phasing of ccsmethphase using the HG002 CCS data.**
**a** Pipeline of ccsmethphase for calling haplotype-aware methylation using CCS data. **b** Distribution of methylation differences of known imprinted intervals calculated using CCS data between two haplotypes of HG002. 96 out of 102 "well-characterized" intervals, and 95 out of 102 "other" intervals which have at least 5 CpGs covered by CCS reads in each haplotype are analyzed. The boxes inside the violin plots indicate 50th percentile (middle line), 25th and 75th percentile (box), the smallest value within 1.5 times interquatile range below 25th percentile and largest value within 1.5 times interquatile range above 75th percentile (whiskers). **c,**

**d** Distribution of the number of BS-seq-generated and ONT-generated DMRs in terms of distance to the closest CCS-generated DMR. **e** Screenshot of Integrative Genomics Viewer (chr20:60,671,001-60,673,750) on a DMR of HG002 near the maternally imprinted gene *GNAS*. Red and blue dots represent CpGs with high and low methylation probabilities, respectively. **f, g** Comparing of PacBio CCS with BS-seq and nanopore sequencing on site-level methylation frequencies of maternal and paternal haplotypes phased by Illumina trio data. Methyl. diff. methylation difference, *r*: Pearson correlation, ONT nanopore sequencing. Source data underlying **b**, **c**, and **d** are provided as a Source Data file.

Using BiGRU and attention mechanism, we designed two deep-learning models in ccsmeth for 5mCpG detection at the read level and the genome-wide site level, respectively. Furthermore, we developed a Nextflow pipeline ccsmethphase for methylation phasing and ASM detection using only PacBio CCS reads.

We systematically evaluated ccsmeth on multiple human samples using controlled (PCR-treated and M.SssI-treated) methylation data-sets, BS-seq and nanopore sequencing. ccsmeth outperforms two existing CNN-based methods on both read-level and site-level 5mC prediction. Evaluation of long CCS reads shows that ccsmeth achieves

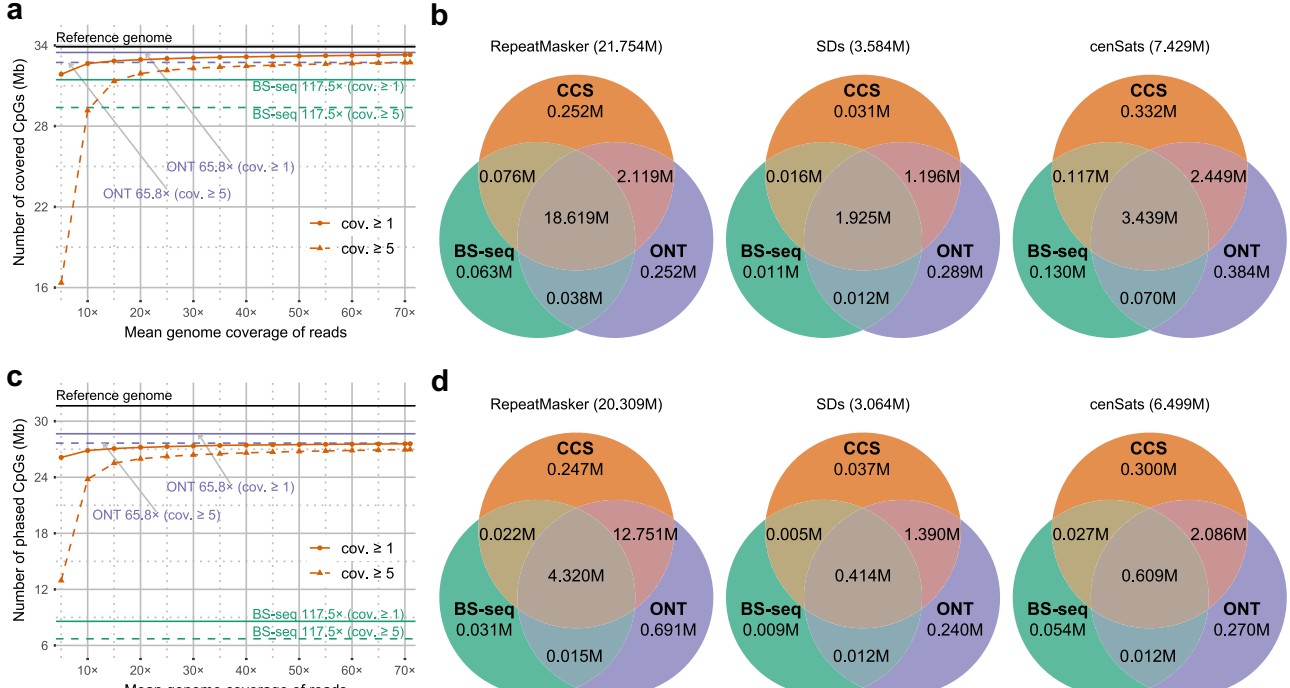

**Fig. 5 | Comparison of the number of CpGs detected/phased by using CCS/BS-seq/nanopore sequencing in the human genome. a** The number of CpGs in autosomes and sex chromosomes detected by using difference coverage of HG002 CCS reads. Values for 5×–70× are the average of 5 repeated tests. **b** Comparison of the number of CpGs detected by the total HG002 BS-seq (117.5×), ONT (65.8×), and CCS (71.0×) reads in repeats annotated by RepeatMasker, segmental duplications, and peri/centromeric regions of autosomes and sex chromosomes. CpGs covered by at least 5 reads are analyzed. **c** The number of CpGs in autosomes phased by using difference coverage of HG002 CCS reads. Values for 5×-70× are the average of 5 repeated tests. **d** Comparison of the number of CpGs phased by using the total HG002 BS-seq (117.5×), ONT (65.8×), and CCS (71.0×) reads in repeats annotated by RepeatMasker, segmental duplications, and peri/centromeric regions of autosomes. CpGs covered by at least 5 phased reads are analyzed. The standard deviation values of the multiple repeated tests of figures **a** and **c** are in Supplementary Tables 14–15. Values in the titles of Venn graphs in sub-figures **b** and **d** are the total number of CpGs in corresponding regions of the T2T-CHM13 genome. cov. coverage, SDs segmental duplications, cenSats peri/centromeric satellites. Source data underlying **a** and **c** are provided as a Source Data file.

better performance on site-level prediction, especially under low coverages. Additionally, we find that the site-level model of ccsmeth can also be applied to the read-level results of other CCS-based methods. The experiments also indicate that with the improvements in read quality and yield (such as the recently introduced Revio system), PacBio CCS has the potential to accomplish genome assembly, detection of SVs, SNVs, and methylation using a single sequencing run of a single sample.

Detecting methylation in non-human species is also important in the field of genetics and epigenetics. In this study, we further performed PacBio CCS and BS-seq of a Zebrafish DNA sample in parallel to evaluate ccsmeth with non-human data (Methods). We used the pretrained model of ccsmeth to detect 5mCpG methylation from the CCS reads of the Zebrafish sample and compare the results with BS-seq. We also detected 5mCpG methylation from the CCS reads using primrose for comparison. The results show that ccsmeth gets higher correlations with BS-seq than primrose does (0.8463 vs. 0.8292), which demonstrates the robustness of ccsmeth for detecting methylation from non-human data (Supplementary Fig. 20).

To evaluate ccsmethphase, we designed the other two pipelines for methylation phasing using BS-seq and nanopore sequencing, respectively, which were referenced from several previous studies[34,58,59]. The results of genome-wide methylation phasing using ccsmethphase are highly consistent with BS-seq and nanopore sequencing, including in known imprinted intervals. Assessment of ccsmethphase shows that PacBio CCS is comparable to nanopore sequencing on methylation phasing. Both of these long-read technologies can detect haplotype-aware methylation states of much more CpGs than BS-seq. While DMRs between haplotypes detected by

nanopore sequencing have been demonstrated to assist haplotyping[48], DMRs detected by PacBio CCS may also be further studied on assisting haplotyping and genome assembly.

Currently, there are also limitations in ccsmeth, as well as other methods using PacBio CCS for methylation detection. First, although PacBio CCS reads can be used to detect strand-specific methylation, all these methods only consider symmetric methylation and combine the features from both DNA strands to predict the methylation states. Hence, these methods are incapable of detecting hemimethylated CpGs, and cannot be applied for 5mC detection in non-CpGs or the detection of other DNA modifications (such as 6mA[60]) which don't have symmetric methylation patterns either. We redesign the model of ccsmeth to call strand-specific methylation using long CCS reads (Supplementary Fig. 21a). The strand-specific-methylation model achieves 0.85 accuracy in the HG002 15 Kb dataset at read level (Supplementary Fig. 21b). However, compared to the symmetric-methylation model, the performance of this model is significantly reduced due to limited subread depth (Supplementary Fig. 21b, c). Second, whether the design of a site-level model can be directly applied to the detection of non-CpG 5mCs and other modifications has not been verified yet. However, by generating more ground-truth datasets for other modifications and re-designing models of ccsmeth according to corresponding patterns of other modifications, we believe the limitations may all be addressed in future research.

In summary, together with PacBio CCS, ccsmeth and ccsmethphase can become well-applicable methods for genome-wide 5mCpG detection and methylation phasing. We expect that our proposed methods will facilitate the analysis of haplotype-aware methylation mechanisms as well as the detection of other modifications.

# Methods

## Ethical statement and sample collection

This study is compliant with the "Guidance of the Ministry of Science and Technology (MOST) of China for the Review and Approval of Human Genetic Resources". The genome sequencing of the Chinese sample SD0651_P1 and the Chinese family trio (HN0641_FA, HN0641_MO, HN0641_S1) was approved by the Research Ethics Committee in the School of Life Sciences, Central South University (No. 2021-1-6). We selected the Chinese samples from the Chinese autism spectrum disorder cohort[61] with no specific sex or age requirements. All the participants signed the informed consent before sample collection. The genomic DNA for PacBio sequencing and bisulfite sequencing was extracted from the peripheral blood of each sample. For the experiment of Zebrafish sample, all animal protocols were reviewed and approved by the Animal Care and Use Committee at Zhongshan Ophthalmic Center, Sun Yat-sen University.

## PacBio CCS data of human

We sequenced a SMRT cell CCS data of NA12898 (GM12898 cell line from Coriell Institute). -11 μg genomic DNA was extracted using QIA-GEN MagAttract HMW DNA Kit (QIAGEN, Cat# 67563). The extracted DNA of NA12898 was amplified via whole genome amplification. Half of the amplified DNA was then treated with the CpG methyltransferase M.SssI. Before library preparation, the genomic DNA was sheared to ~20 Kb on a MegaRuptor3 (Diagenode). Libraries of the M.SssI-treated DNA and the other half amplified DNA were prepared in 10 Kb insert size with the Express Template Prep kit 2.0 (PacBio, No. 100-938-900), and were then barcoded to sequence on a PacBio Sequel II sequencer with Sequel II sequencing kit 2.0 (PacBio, No. 101-826-100). We sequenced the native genomic DNA of four Chinese samples using the same procedure on the Sequel II system. We got 2 SMRT cells of CCS reads (19.6× mean genome coverage in total) in 15Kb insert size for SD0651_P1. We also got 2 SMRT cells of CCS reads in 15 Kb insert size for the HN0641 trio samples. There are 21.8×, 22.4×, and 21.1× reads for HN0641_FA, HN0641_MO, and HN0641_S1, respectively.

We downloaded three CCS raw subreads of a human sample from Tse et al.[24]: M01 and W01; M02 and W02; M03 and W03. Each of the three datasets contains two groups of reads: the methylated reads sequenced using M.SssI-treated DNA (M01, M02, and M03) and the unmethylated reads sequenced using amplified DNA (W01, W02, and W03). The three datasets were sequenced on PacBio sequencers with Sequel I sequencing kit 3.0, Sequel II sequencing kit 1.0, and Sequel II sequencing kit 2.0, respectively.

We downloaded 9 SMRT cells of HG002 CCS raw subreads in 15 Kb, 20 Kb, and 24 Kb insert sizes from Baid et al.[26] and Human Pangenome Reference Consortium[23]. We also got 2 SMRT cells of CHM13 CCS raw subreads in 20 Kb insert size from Nurk et al.[27]. CCS subreads in all datasets were processed to generate CCS reads using pbccs (v6.4.0, https://github.com/PacificBiosciences/ccs). Details of all CCS data are provided in Supplementary Table 1.

## Data partition of the PacBio CCS data of human

For each dataset of M.SssI-treated and PCR-treated human DNA from Tse et al.[24], we randomly select 50% methylated reads and 50% unmethylated reads for model training, while the remaining 50% methylated and unmethylated reads are used for evaluation at read level.

To evaluate ccsmeth using datasets of long CCS reads, we train the read-level model of ccsmeth using NA12898 CCS reads aligned to autosomes and a SMRT cell of HG002 CCS reads. We use another 2 SMRT cells of HG002 reads for site-level model training. The CCS reads of NA12898 aligned to chrX and chrM, the left 6 SMRT cells of HG002 CCS reads, the 2 SMRT cells of CHM13 CCS reads, and the CCS reads of the HD0641 family trio are used for evaluation (Supplementary Table 2).

## Illumina and nanopore data of human

We sequenced the SD0651_P1 sample using BS-seq. The extracted genomic DNA (≥1 μg) was first sheared by Covaris and purified to 200–350 bp. The sheared DNA was then end-repaired and ligated to methylated adapters. The adapter-ligated DNA was bisulfite-converted with EZ DNA Methylation-Gold Kit (Zymo Research, Cat# D5006) and then PCR-amplified. Qubit® 2.0 Fluorometer (Invitrogen) was used to quantify the DNA fragments of the library. Finally, the library was sequenced on a NovaSeq6000 sequencer (Illumina). In total, we got 15.7× coverage of 2 × 150 bp paired reads.

We downloaded BS-seq and nanopore R9.4.1 data of HG002 from ONT Open Datasets (https://labs.epi2me.io/dataindex/). There are 117.5× coverage of 2 × 150 bp paired reads of BS-seq, and 9.5 million (65.8× coverage) nanopore reads with a mean length of 21,933 bp for HG002. We also got Illumina whole-genome sequencing (WGS) trio data of HG002 from GIAB[62], in which there are 63.1×, 55.7×, and 67.9× coverage of 2 × 250 bp paired reads for HG002, HG003, and HG004, respectively. We downloaded 6.7 million (41.8× coverage) nanopore R9.4.1 reads of CHM13 from Nurk et al.[27]. The mean length of the CHM13 nanopore reads is 19,891 bp.

We used Bismark[63] (v0.23.1) to process all BS-seq data. For the nanopore data, we basecalled the Fast5 files (raw reads) using Guppy (version 4.2.2+effbaf8). Then we used DeepSignal2 (v0.1.2, https://github.com/PengNi/deepsignal2), an improved version of DeepSignal[17], to call methylation from the nanopore reads. Details of all Illumina and nanopore data used in this study are provided in Supplementary Table 3.

## Reference genome and annotations

We used CHM13 v2.0[27] as the human reference genome to process all sequencing data. The gene annotations were downloaded from the GitHub repository marbl/CHM13[27]. The annotations of repetitive genomic elements (RepeatMasker)[54,55], segmental duplications[56], peri/centromeric satellites[57], and CpG islands were downloaded from corresponding tracks of UCSC Genome Browser (T2T CHM13v2.0/hs1)[64].

We got 205 known imprinted intervals of human from Akbari et al.[48], which were generated from five previous studies[49–53]. Of these intervals, there were 102 "well-characterized" intervals that were reported by at least two studies[49]. By using UCSC LiftOver[65], GRCh38 coordinates of 204 intervals (102 "well-characterized" and 102 "other" intervals) were successfully converted to CHM13 coordinates.

## PacBio CCS and BS-seq data of Zebrafish

We sequenced the Zebrafish sample using PacBio CCS and BS-seq with the same procedure for sequencing the human samples. The genomic DNA of Zebrafish was extracted from the muscular tissues of the Zebrafish adults (TU wild-type line, male and female), which were provided by China Zebrafish Resource Center. >10 μg and >1 μg DNA was used for PacBio CCS and BS-seq, respectively. In total, we got 23.3× CCS reads and 29.5× BS-seq reads.

## Methylation calling of ccsmeth at read level

To call CpG methylation at read level, ccsmeth needs CCS reads with kinetics information in BAM format, which can be generated from raw subreads by pbccs with "--hifi-kinetics" option. The process of ccsmeth to call methylation at reads level is as follows (Fig. 1):

(1) Feature extraction. Each CCS read with kinetics information contains IPD and PW values for bases in forward and reverse complement strands of the read, which are averaged from corresponding bases in subreads. Before extracting features for CpGs in a CCS read, we first normalize the IPD and PW values of each strand in the read using Z-score normalization. Then for a CpG in the forward strand of the CCS read, we extract a 21-mer sequence context surrounding the CpG. Finally, the averaged IPD and PW values, the number of covered subreads of each base in the 21-

mer, together with the 21-mer nucleotide sequence form a $4 \times 21$ feature matrix. We also construct a feature matrix for the symmetric CpG in the reverse complement strand using the same way.

(2) Methylation state prediction. We use the two feature matrixes to predict a single methylation state for the symmetric CpG pair. Each of the two matrixes is fed into a deep neural network, which contains three bidirectional Gated Recurrent Unit (BiGRU) layers[42] and one Bahdanau attention layer[43] (Fig. 1b, Supplementary Note 5). Each BiGRU layer has a hidden size of 256. The outputs from the two attention layers are processed by a full connection layer and then by a Softmax layer. Finally, a methylation probability $P_r$ is outputted. A binary methylation state of the CpG is also set based on $P_r$: if $P_r > 0.5$, the CpG is predicted as methylated, otherwise is predicted as unmethylated.

## Methylation calling of ccsmeth at site level

We used the following steps to call CpG methylation at site level:

(1) Alignment. CCS reads should be aligned to the reference genome before site-level methylation calling. We use pbmm2 (v1.9.0, https://github.com/PacificBiosciences/pbmm2), a modified version of minimap2[66] for PacBio native data formats, to align all the CCS reads used in this study.

(2) Methylation calling in count mode. In count mode, based on the binary methylation states of CpGs in the mapped reads, the methylation frequency of a CpG is calculated as the number of reads where the CpG is called methylated divided by the total number of reads mapped to the CpG (Supplementary Fig. 1).

(3) Methylation calling in model mode. In model mode, we used the information of neighboring CpGs to predict site-level methylation frequencies in a way similar to pb-CpG-tools. For a targeted CpG, the read-level methylation probabilities of the CpG and each of its 10 adjacent CpGs are first summarized in a histogram with 20 discretized bins separately. Then each histogram is normalized by its L2-norm[67] value. The distances of 11 CpGs to the targeted CpG have also been calculated. 11 normalized histograms and the 11 distance values are organized into a $21 \times 11$ feature matrix, which is then fed into a BiGRU layer and an attention layer (Fig. 1c). The BiGRU layer has a hidden size of 32. At last, ccsmeth outputs a methylation probability $P_s$ ($P_s \in [0,1]$) as the methylation frequency of the targeted CpG.

## Model training of ccsmeth

(1) Training of the read-level model. For the datasets of M.SssI-treated and amplified DNA, we extract positive (methylated) and negative (methylated) samples from reads of M.SssI-treated and PCR-treated DNA, respectively. For CCS data of native DNA, based on the results of BS-seq, we take CpGs which are covered with at least 5 reads and have 100% methylation frequency as high-confidence methylated sites. CpGs that have at least 5 mapped reads and zero methylation frequency are selected as high-confidence unmethylated sites. Then we extract positive and negative samples from the reads which are mapped to the high-confidence methylated and unmethylated sites, respectively.

To train the model of ccsmeth for read-level 5mCpG detection, we split the total training samples at a ratio of 99:1 as the training dataset and the validation dataset. The model parameters are learned on the training dataset by minimizing the loss calculated by cross-entropy (Supplementary Note 5) with a batch size of 512 and an initial learning rate of 0.001. The learning rate is adopted by Adam optimizer[68] and decays by a factor of 0.1 after every epoch. The parameter betas in Adam optimizer are set to (0.9, 0.999). We use two strategies to prevent overfitting. First, we add a dropout layer in each of the GRU layers and the fully connected

layer. We set the dropout probability to 0.5 at each dropout layer. Second, we use early stopping[69] during training. We set at least 10 epochs and at most 50 epochs for each training. The model parameters with the current best performance on the validation dataset are saved in every epoch. During epochs 10 to 50, if the best performance of the current epoch decreases, we stop the training process.

(2) Training of the site-level model. The training of the site-level model of ccsmeth is treated as a regression problem. From the results of BS-seq, we select CpGs with at least 10× coverage, and use the methylation frequencies of the CpGs as training targets. We then generate the read-level methylation probabilities calculated by ccsmeth for each targeted CpG as features. The targeted CpGs, alongside the features, are then split at a ratio of 99:1 as the training dataset and the validation dataset. Finally, we use the same training process of the read-level model to train the site-level model but with a different loss function: during the training of the site-level model, we use the mean squared error (MSE) (Supplementary Note 5) instead of cross-entropy to calculate the loss.

## Evaluation of ccsmeth

(1) Evaluation at read level. For the controlled methylation datasets, we extract positive and negative samples from the reads of PCR-treated and M.SssI-treated DNA, respectively. For the CCS reads of HG002 and SD0651_P1, we first select high-confidence methylated and unmethylated sites from the results of BS-seq. Then we extract positive and negative samples of the selected sites from CCS reads. We calculate accuracy, sensitivity, specificity, and Area Under the Curve (AUC) based on the prediction of randomly selected 100,000 positive samples and 100,000 negative samples. We repeat the subsampling 5 times for each evaluation. (2) Evaluation at site level. We evaluate ccsmeth at the genome-wide site level by comparing per-site methylation frequencies predicted by ccsmeth with the results of BS-seq and nanopore sequencing. Using the methylation frequencies of CpGs, Pearson correlation ($r$), the coefficient of determination ($r^2$), Spearman correlation ($\rho$), and root mean square error (RMSE) are calculated. For each comparison, we only compare CpGs that have at least 5× coverage in results of both PacBio CCS and BS-seq (or nanopore sequencing). To evaluate the methylation frequencies predicted by ccsmeth under different coverage of CCS reads, we randomly subsample the CCS reads using rasusa[70] (v0.7.0).

ccsmeth is implemented using Python3 and PyTorch (version 1.11.0). We evaluate ccsmeth, HK model, primrose (version 1.3.0), and pb-CpG-tools (v1.1.0) on the same testing datasets. The source code of the HK model was taken from Tse et al.[24] under the CUHK software license. We also compared the runtime and peak memory of the main steps of ccsmeth, HK model, and primrose (Supplementary Note 7, Supplementary Fig. 22, and Supplementary Tables 18–19).

## ccsmethphase for methylation phasing and ASM detection

In the ccsmethphase pipeline (Fig. 4a), we use ccsmeth to call read-level methylation from PacBio CCS reads. Then the CCS reads are aligned to the reference genome by using pbmm2 (v1.9.0). We use Clair3[33] (v0.1-r11 minor 2) with the "*hifi*" model to call variants and only keep the "PASS" SNVs (*i.e.*, high-quality SNVs). We use WhatsHap[46] (version 1.4) to phase the "PASS" SNVs, and then to phase the reads (i.e., tag the reads by the haplotypes). The methylation frequencies of CpGs in each haplotype are then inferred by ccsmeth. At last, we use DSS[47] (version 2.44.0) with default parameters to perform differential methylation analysis (DMA). By taking methylation frequencies of CpGs in two haplotypes as input, DSS calls differentially methylated regions (DMRs) using Wald tests[47]. We only consider regions generated

by DSS with $p$-value < 0.001 and |methylation difference| > 0.2 as significant DMRs.

ccsmethphase is implemented by integrating ccsmeth and other necessary tools using Nextflow (version 22.04.5.5708). We evaluated the runtime of the ccsmethphase pipeline on an HPC cluster. Details of the evaluation are shown in Supplementary Note 7, Supplementary Table 20, and Supplementary Fig. 23.

## Methylation difference of known imprinted intervals between two haplotypes

We compare the haplotype-level methylation difference of each known imprinted interval calculated by ccsmethphase with the results of BS-seq and nanopore sequencing (Supplementary Note 3 and 4). For each interval, we first split the CpGs in the reads which are mapped to the interval into two groups ($G_{hp1}$ and $G_{hp2}$) according to the haplotype tags of the reads. Then we calculate the methylation level of the interval in each haplotype ($M_{hp1}$ and $M_{hp2}$) as the fraction of CpGs that are predicted as methylated in the corresponding group (Eq. (1)). At last, we calculate the methylation difference of the interval between two haplotypes (Eq. (2)). Note that we only calculate the methylation difference of intervals that have at least 5 CpGs covered by reads in both haplotypes.

$$M_{hp1} = \frac{\text{No. of methylated CpGs in } G_{hp1}}{\text{No. of total CpGs in } G_{hp1}}, M_{hp2} = \frac{\text{No. of methylated CpGs in } G_{hp2}}{\text{No. of total CpGs in } G_{hp2}} \quad (1)$$

$$M_{diff} = |M_{hp1} - M_{hp2}| \quad (2)$$

## Statistics and reproducibility

This study obtained 5 human samples (NA12898, SD0651_P1, HN0641_FA, HN0641_MO, HN0641_S1) for generating PacBio CCS data. We also used publicly available datasets of samples M01&W01, M02&W02, M03&W03, HG002, and CHM13. The training and evaluation process of the proposed method followed standard practices for separating training, validation, and testing datasets. During evaluation, we performed orthogonal validation for samples HG002, CHM13, and SD0651_P1 by using nanopore sequencing and bisulfite sequencing. We also compared the results of sequencing data from different HG002 SMRT cells to validate the reproducibility. No statistical method was used to predetermine sample size. CCS reads that have less than 3 full-length subreads, and "Fail" Nanopore sequencing reads (mean Q-score ≤ 9) were not used in the study. The experiments were not randomized. The Investigators were not blinded to allocation during experiments and outcome assessment. All the statistical details for training and evaluation can be found in the figure legends, Methods section, and Supplementary information.

## Reporting summary

Further information on research design is available in the Nature Portfolio Reporting Summary linked to this article.

## Data availability

All sequencing data generated in this study have been deposited in the Genome Sequence Archive[71] in National Genomics Data Center[72], Beijing Institute of Genomics, BIG, http://gsa.big.ac.cn), Chinese Academy of Sciences, under Project accession No. PRJCA015556. The sequencing data of NA12898 (GSA-Human accession No. HRA004180) and the Zebrafish sample (GSA accession No. CRA010412) is available under open access. The sequencing data of SD0651_P1 and the HN0641 family trio (GSA-human accession No. HRA004202) is available under restricted access, which can be granted by the Data Access Committee (DAC). Access can be obtained for research use only by completing the application form via GSA. Users can register and login to GSA [https://

ngdc.cncb.ac.cn/gsa-human/] and follow the guidance of "Request Data" [https://ngdc.cncb.ac.cn/gsa-human/document/GSA-Human_Request_Guide_for_Users_us.pdf] to request the data.

The CCS datasets of M.SssI-treated and PCR-treated DNA (M01-03, W01-03) are available from Tse et al.[24]. The CCS reads of HG002 are available from Google Cloud[26] [https://console.cloud.google.com/storage/browser/brain-genomics-public/research/deepconsensus/publication/sequencing] and the Human Reference Pangenome Consortium GitHub repository [https://github.com/human-pangenomics/HG002_Data_Freeze_v1.0]. Raw nanopore reads of HG002 are available at ONT Open Datasets [https://labs.epi2me.io/gm24385_2020.11/] with the flowcell ID PAG07165. The BS-seq reads of HG002 are also available at ONT Open Datasets [https://labs.epi2me.io/gm24385-5mc/]. The Illumina WGS 2 × 250 bp reads of AshkenazimTrio (HG002, HG003, and HG004) are available at the GIAB GitHub repository [https://github.com/genome-in-a-bottle/giab_data_indexes]. The CHM13 CCS and nanopore reads are available at GitHub repository marbl/CHM13[27] [https://github.com/marbl/CHM13]. Source data are provided with this paper.

## Code availability

ccsmeth is publicly available at GitHub [https://github.com/PengNi/ccsmeth]. ccsmethphase is publicly available at GitHub [https://github.com/PengNi/ccsmethphase] and Zenodo[73].

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

## Acknowledgements

This work was supported in part by the National Key Research and Development Program of China (No. 2021YFF1201200); the National Natural Science Foundation of China under Grants (Nos. 62350004, 62150048, U1909208); the Open Project of Xiangjiang Laboratory (Nos. 22XJ02002, 22XJ03010) to Jianxin Wang. This work was also supported in part by the US National Institute of Food and Agriculture (NIFA; Grant Number 2017-70016-26051) and the US National Science Foundation (NSF; Grant Number ABI-1759856, MRI-2018069, MTM2-2025541) to Feng Luo. This work makes use of the program (for the HK model) and data generated by The Chinese University of Hong Kong (CUHK) Department of Chemical Pathology, as reported by Tse et al., of whom we are very thankful, in Proc Natl Acad Sci USA 2021; 118(5): e2019768118. We thank Oxford Nanopore Technologies, Baid et al. and Human Pangenome Reference Consortium, the GIAB team, and Nurk et al. for making the HG002 nanopore and BS-seq data, the HG002 CCS data, the AshkenazimTrio Illumina data, and the CHM13 CCS and nanopore data publicly available, respectively. We also thank Nicole Newell and Aaron Wenger from PacBio for useful discussions. This work was carried out in part using computing resources at the High-Performance Computing Center of Central South University.

## Author contributions

J.X.W. and F.L. conceived and designed this project. P.N., J.X.W., and F.L. conceived, designed, and implemented the ccsmeth and ccsmethphase. F.N. helped design ccsmeth and the pipeline of ccsmethphase. C.L.X. helped sequence the PacBio CCS and BS-seq reads of the NA12898 and Zebrafish DNA samples. J.C.L. and Y.F.H provided the DNA samples of SD0651_P1 and the HN0641 trio, and helped sequence the PacBio CCS and BS-seq reads. P.N., F.N., Z.Y.Z., J.R.X., N.H., and J.Z. evaluated ccsmeth and ccsmethphase using the sequencing data. H.C.Z helped train the site-level model of ccsmeth. Y.Z. helped process the sequencing data on the HPC cluster. P.N., F.L., and J.X.W. wrote the paper. All authors have read and approved the final version of this paper.

## Competing interests

The authors declare no competing interests.
