## [Peer Review File · Nature Communications]

DNA 5-methylcytosine detection and methylation phasing using PacBio circular consensus sequencingReviewer #1 (Remarks to the Author):

The authors develop a method to identify CpG methylation using PacBio CCS reads. Direct methylation detection using sequencing characteristics rather than bisulfite conversion or similar alternatives is an area of active research. Any progress would be valuable. The authors find 90% accuracy in methylation calls using reads longer than 10kb. The authors do a good job at describing the current state of the field and cite important alternatives to their own approach.

They evaluate the results of their analysis using correlations with BS-seq and Oxford Nanopore sequencing results. They also explore the practical consequences of their methodology using a DMR analysis.

The tool presented by the authors appears to be an incremental improvement, with a notable decrease in the read depth required for an accurate methylation call. The comparisons made by the authors to currently available alternatives seem fair and comprehensive.

Some questions remain, specifically whether the methylation call errors are correlated with any genomic features or if there are any potential biases in the accuracy rate. These are likely questions for future studies.

Minor comments:

It is possible to determine from context, but a clarification of "read level" vs "site level" calls would be useful.

Page 14 "closely next to": is this referring to genomic location? Should be clarified.

Reviewer #2 (Remarks to the Author):

Ni et al. describe ccsmeth and ccsmethphase, a set of software packages for extracting and analyzing DNA methylation data from Pacific Biosciences long read sequences. Overall, I think this is an important new toolset, given the high degree of interest in PacBio sequencing and the ability to analyze DNA methylation information. While the ability to analyze methylation from PacBio reads has been described last year, this manuscript is important for several reasons: (1) the other two competing tools, "HK model" and "primrose", are either not publicly available or only available as an executable - neither provides source code. Ccsmeth is open-source, and a complete reproducible and portable pipeline is provided as a Nextflow workflow. (2) There has to date been no systematic benchmarking of PacBio methylation calling compared to bisulfite-seq or native Oxford Nanopore methylation, and this manuscript provides benchmarking of these relative to both ccsmeth and the other two competing PacBio methods. (3) A full pipeline to phase methylation vs. parent-of-origin haplotypes, along with internal validation and comparison of phasing using bs-seq and Oxford Nanopore, is especially important given that this is one of the main reasons to use the PacBio platform, which provides both long reads and high accuracy base calls necessary for haplotype phasing.

Despite my enthusiasm for this work, I do have a few concerns which should be addressed.

I don't understand why CCS can only detect symmetric methylation. From the illustration in Figure 1, it looks like you get independent subreads from the forward and reverse strands. Is there something inherent to the technology why you can't call the cytosine on each strand independently? If so, please address this in the introduction or the discussion. This would be a major advantage to the PacBio platform, since true single-molecule methylation on both strands is difficult or impossible to achieve with other technologies, and we therefore lack knowledge about hemimethylation. Oxford Nanopore demonstrated internal results on this recently with their "duplex" sequencing, so it will be important for the audience to understand if it's possible on PacBio.

One of the important aspects for end users is computational efficiency. How long does it take to run, what kind of hardware is required, etc. I realize that you show some of this information in Supp Note 5, but it's not sufficient. You should have a table comparing both hardware requirements, as well as run-time, with HK model (if available) and Primrose.

I don't understand how you ran the HK model for evaluation. I can not find an executable version available online, and your Methods section does not give any details on the HK model software that you ran (does it have a version number? Where is it downloaded from?). You should also add this information to the "Software and Code" section of your Nature portfolio Reporting Summary.

On p.10 You say you could not benchmark HK model on NA12898,HG002,SD0651 because "HK model is extensively time-consuming on large datasets". I am concerned that you do not provide HK model benchmarking for these cell types. Please give more details for my review on which step takes too long to run, how long it takes, does it require too much RAM, etc. I need to understand why you are not able to do this. Why can't you just use a small subset of randomly sampled reads for evaluation, if it's not possible to run on the full datasets?

I have worked extensively on whole-genome methylation (both WGBS and Nanopore), and I am perplexed about why the "model mode" performs so much better than the "count mode" for site-level methylation calls. I could understand if this was the result of low sequence coverage, but the "count mode" doesn't seem to equalize with the "model mode" even at 70x sequencing depth (Supplemental Figure 2). Since you advocate for the "model mode", I think you should try to address this in the Discussion, or perform additional analyses to understand what CpG contexts are significantly improved under the "model mode". I do have a concern about overfitting specific cell types or data types in the training data, but maybe this is addressed in the CHM13 sample, which is not included in the training dataset?

For count mode, why do you include very ambiguous calls (those with probability close to 0.5)? I think you would have better accuracy if you filtered out ambiguous reads (for instance, reads with probability 0.33-0.66, as the Oxford Nanopore modbam2bed tools does) <https://github.com/epi2me-labs/modbam2bed> . I would like to see a new version of Fig 3C-D which uses such a filtering, just to see if it improves the accuracy.

Data availability section states, "The sequencing data of NA12898, SD0651_P1 and the HN0641 family trio will soon be available at NCBI." It is important that the reviewers can see these datasets (even if they are available). It's also important to have the methods described (how patient samples were collected and processed, etc.)

Minor

For Fig 3A-D, could you use the same Y axis limits for all plots? That would make it easier to compare.

In Fig 5B and 5D, it would be good to show a Venn Diagram for the non-repeatmasker (unique) CpGs as well.

p.9: Grammatical fix. "are got from Baid et al" should be "were taken from Baid et al"

p.9: You start by saying that these experiments used NA12898. But then several times you reference NA12878. Was this a typo, or are you actually using both of these two samples? It was not clear.

Ben Berman
Hebrew University of Jerusalem, Israel

Reviewer #3 (Remarks to the Author):

Ni et al. present the work of ccsmeth for identifying CpG methylation using PacBio CCS data and ccsmethphase for phasing methylation and identifying differentially methylated regions between haplotypes. Overall, the methods are of great interest to the community, the data appear to be high quality, and the results are well presented. Specifically, they show that ccsmeth performs better than primrose, another CCS methylation calling program developed by PacBio, and is comparable to nanopore data called by deepsignal2. They also applied the method across different human samples and showed good correlations with the traditional BS-seq data. These are very exciting results for human studies.

One major concern is the generalizability of the research and the potential for overtraining. For the potential readership of Nature Communications, a large number of researchers are studying non-human non-model species that lack proper training data as their research targets, such as fungi, plants, and non-human vertebrates. It is important to document to what extent their model can be applied to other species, especially since nanopore approaches and even primrose have been shown to be robust with non-human data.

Minor:

1. Page 6, paragraph 2, the authors claimed ccsmethphase could be the first haplotype-aware methylation calling tool. This may not be true. For example, modbamtools (<https://rrazaghi.github.io/modbamtools/>) developed by Razaghi et al. (2022), can identify and visualize CpG haplotypes.
2. End of page 7 - beginning of P8, "Then the feature matrix is fed into one BiGRU layer and then into one attention layer" could benefit from more context and details.
3. Figure 1a, the orange bars are not labeled.
4. Figure 3f, the correlation between ccsmeth and BS-seq in the SD0651_P1 sample is relatively low ($r = 0.8750$) compared to HG002 (Figure 3e, $r \geq 0.9271$), can you comment on this?
5. Suppl. Fig. 9a, can you comment on why ccsmeth identified DMRs are more distant away from the BS-seq identified DMRs?
6. For tests involving multiple replications, plotting the standard deviation should be helpful to contextualize the variation in program performance. For example, Fig2, Fig3a-d, Fig 5ac, and several suppl Figs.

Summary

We appreciate the valuable comments and suggestions from the editor and reviewers. Based on the suggestions and comments from the editor and reviewers, we revised our paper. We addressed those comments and suggestions carefully and included a point-by-point response below. The significant changes in the revised manuscript were highlighted in red color.

Response to Reviewer #1

Reviewer #1 (Remarks to the Author):

The authors develop a method to identify CpG methylation using PacBio CCS reads. Direct methylation detection using sequencing characteristics rather than bisulfite conversion or similar alternatives is an area of active research. Any progress would be valuable. The authors find 90% accuracy in methylation calls using reads longer than 10kb. The authors do a good job at describing the current state of the field and cite important alternatives to their own approach.

They evaluate the results of their analysis using correlations with BS-seq and Oxford Nanopore sequencing results. They also explore the practical consequences of their methodology using a DMR analysis.

The tool presented by the authors appears to be an incremental improvement, with a notable decrease in the read depth required for an accurate methylation call. The comparisons made by the authors to currently available alternatives seem fair and comprehensive.

We thank the reviewer for the positive comments.

Some questions remain, specifically whether the methylation call errors are correlated with any genomic features or if there are any potential biases in the accuracy rate. These are likely questions for future studies.

Thanks for the above concern and suggestion! To test whether the methylation call errors are correlated with any genomic features, we evaluate ccsmeth for read-level 5mCpG detection in the following genomic contexts and regions: (1) Singletons and non-singletons; (2) Genic regions (promoters, exons, introns) and intergenic regions; (3) CpG islands, shores, and shelves; (4) Repetitive regions which include simple repeats, short interspersed nuclear elements (SINE), long interspersed nuclear element (LINE), long terminal repeat (LTR), and other repetitive regions. We find that ccsmeth tends to have higher performances in regions with high CpG densities, such as non-singletons and CpG islands. ccsmeth has relatively lower accuracies in intergenic regions, CpG shores, and CpG shelves. In simple repeats and “Others” repetitive regions, ccsmeth has lower sensitivities and specificities, respectively. The results show that biologically relevant genomic contexts and regions do impact the performance of 5mCpG detection. However, as shown in Supplementary Fig. 3, ccsmeth outperforms primrose in all tested regions.

We have added this analysis in detail in Supplementary Note 1, Supplementary Fig. 3, and the *Results* section in the revised manuscript.

Minor comments:

It is possible to determine from context, but a clarification of "read level" vs "site level" calls would be useful.

Thanks for the above suggestion. We have added a clarification for “read-level” and “site-level” prediction in the *Results – The ccsmeth algorithm for 5mCpG detection* section is as follows:

“For a targeted CpG, ccsmeth first predicts the methylation probability (or binary methylation state) of the CpG in a read (*i.e.*, at single-molecule resolution, read level), and then summarize the read-level methylation states to get its methylation frequency(site-level) in the targeted genome.”

Page 14 "closely next to": is this referring to genomic location? Should be clarified.

Yes, this is referring the genomic locations of the DMRs. We have made this clearer in the revised manuscript.

Response to Reviewer #2

Reviewer #2 (Remarks to the Author):

Ni et al. describe ccsmeth and ccsmethphase, a set of software packages for extracting and analyzing DNA methylation data from Pacific Biosciences long read sequences. Overall, I think this is an important new toolset, given the high degree of interest in PacBio sequencing and the ability to analyze DNA methylation information. While the ability to analyze methylation from PacBio reads has been described last year, this manuscript is important for several reasons: (1) the other two competing tools, “HK model” and “primrose”, are either not publicly available or only available as an executable - neither provides source code. Ccsmeth is open-source, and a complete reproducible and portable pipeline is provided as a Nextflow workflow. (2) There has to date been no systematic benchmarking of PacBio methylation calling compared to bisulfite-seq or native Oxford Nanopore methylation, and this manuscript provides benchmarking of these relative to both ccsmeth and the other two competing PacBio methods. (3) A full pipeline to phase methylation vs. parent-of-origin haplotypes, along with internal validation and comparison of phasing using bs-seq and Oxford Nanopore, is especially important given that this is one of the main reasons to use the PacBio platform, which provides both long reads and high accuracy base calls necessary for haplotype phasing.

We thank the reviewer for the positive and supportive comments.

Despite my enthusiasm for this work, I do have a few concerns which should be addressed.

I don't understand why CCS can only detect symmetric methylation. From the illustration in Figure 1, it looks like you get independent subreads from the forward and reverse strands. Is there something inherent to the technology why you can't call the cytosine on each strand independently? If so, please address this in the introduction or the discussion. This would be a major advantage to the PacBio platform, since true single-molecule methylation on both strands is difficult or impossible to achieve with other technologies, and we therefore lack knowledge about hemimethylation. Oxford Nanopore demonstrated internal results on this

recently with their “duplex” sequencing, so it will be important for the audience to understand if it’s possible on PacBio.

Thanks for the above concern! We agree that it is important for PacBio CCS to have the ability to detect strand-specific methylation. And as the subreads from the forward and reverse strands are independent, PacBio CCS can be used to detect strand-specific detection. We perform an analysis to explore the ability of PacBio CCS for detecting single-stranded methylation. We have added this analysis in Supplementary Fig. 21.

As shown in Supplementary Fig. 21a, we redesign the model and the input of ccsmeth to call CpG methylation on each strand independently. While the ccsmeth model for symmetric methylation calling combines signals from subreads of both strands (Fig. 1b), the model for strand-specific methylation calling only uses about half of the subreads from a CCS read for prediction on each strand. We use the long CCS reads (of NA12898, HG002, SD0651_P1) to compare the two ccsmeth models for read-level prediction. As shown in Supplementary Fig. 21b, the strand-specific-methylation model achieves >0.85 accuracies (in the HG002 15Kb dataset) at the read level. However, compared to the symmetric-methylation model, the accuracies of this model are significantly reduced (~5-7%) due to limited subread depth. Supplementary Fig. 21c shows that higher subread depth increases accuracy.

We have also rewritten the corresponding part in the *Discussion* section. In future research, we will study how to use long CCS reads to call strand-specific methylation in both CpGs and non-CpGs more accurately.

One of the important aspects for end users is computational efficiency. How long does it take to run, what kind of hardware is required, etc. I realize that you show some of this information in Supp Note 5, but it’s not sufficient. You should have a table comparing both hardware requirements, as well as run-time, with HK model (if available) and Primrose.

Thanks for the concern and suggestion. In the revised manuscript, we compared the hardware requirements, runtime, and peak memory of the main steps of ccsmeth, HK model, and primrose as suggested.

First, we illustrated the main steps of the three methods for methylation calling in Supplementary Fig. 22.

Then, we subsampled 100K ZMW reads from each of the three HG002 CCS datasets (15Kb, 20Kb, 24Kb) to compare all three methods. We further used 1 SMRT cell CCS reads from each of the three HG002 datasets to compare primrose and ccsmeth. The comparison was performed at an HPC cluster containing two kinds of servers: (1) Server-CPU with 48 CPU cores (Intel(R) Xeon(R) Gold 6248R CPU @ 3.0GHz) and 192 GB RAM; (2) Server-GPU with 40 CPU cores (Intel(R) Xeon(R) Gold 6248 CPU @ 2.50GHz), 384 GB RAM, and 2 Nvidia Tesla V100 GPU cards. The hardware requirements, runtime, and peak memory for each step of the three methods were listed in Supplementary Tables 18-19.

We changed Supplementary Note 5 to Supplementary Note 7 in the revised manuscript and included this analysis in detail in Supplementary Note 7.

I don’t understand how you ran the HK model for evaluation. I can not find an executable version available online, and your Methods section does not give any details on the HK model software that you ran (does it have a version number? Where is it downloaded from?). You should also add this information to the “Software and Code” section of your Nature portfolio Reporting Summary.

We got the source code of the HK model from the corresponding author of the paper [1] by email. HK model does not have a version number. The source code of the HK model can be used for academic non-commercial

research purposes under the CUHK software license. We have made this clearer in the *Methods* section of the revised manuscript and the Reporting Summary file.

References

[1] OY Olivia Tse, Peiyong Jiang, Suk Hang Cheng, Wenlei Peng, Huimin Shang, John Wong, Stephen L. Chan et al. "Genome-wide detection of cytosine methylation by single molecule real-time sequencing." *Proceedings of the National Academy of Sciences* 118, no. 5 (2021): e2019768118.

On p.10 You say you could not benchmark HK model on NA12898,HG002,SD0651 because “HK model is extensively time-consuming on large datasets”. I am concerned that you do not provide HK model benchmarking for these cell types. Please give more details for my review on which step takes too long to run, how long it takes, does it require too much RAM, etc. I need to understand why you are not able to do this. Why can’t you just use a small subset of randomly sampled reads for evaluation, if it’s not possible to run on the full datasets?

Thanks for the above concern and suggestion. In the revised manuscript, we have included the evaluation of the HK model using subsampled long CCS reads as suggested.

The analysis we performed for computational efficiency comparison (as mentioned above) shows that the “*Extract features*” step of the HK model takes too much time to run. As shown in Supplementary Table 15, the “*Extract features*” step needs ~13-19 hours to extract features from datasets containing 100K ZMW reads. This is mainly because the HK model directly extracts features from subreads, which increases the computational complexity compared to csmeth and primrose. And the script of the HK model for “*Extract features*” is not optimized for parallel processing.

To evaluate the HK model on long CCS reads, we first re-trained the HK model using the samples extracted from the same datasets for training csmeth. Then, we subsampled 100K ZMW reads from the datasets of NA12898 and HG002 (15Kb, 20Kb, 24Kb) to test the HK model at the read level. We could not use the SD0651_P1 sample for evaluation because we don’t have the raw subreads of SD0651_P1. The results show that the HK model achieves similar accuracies as primrose, especially on the HG002 datasets. Accuracies of both the HK model and primrose are lower than that of csmeth.

We have added this analysis in Supplementary Fig. 2 and the corresponding *Results* Section in the revised manuscript.

I have worked extensively on whole-genome methylation (both WGBS and Nanopore), and I am perplexed about why the “model mode” performs so much better than the “count mode” for site-level methylation calls. I could understand if this was the result of low sequence coverage, but the “count mode” doesn’t seem to equalize with the “model mode” even at 70x sequencing depth (Supplemental Figure 2). Since you advocate for the “model mode”, I think you should try to address this in the Discussion, or perform additional analyses to understand what CpG contexts are significantly improved under the “model mode”. I do have a concern about overfitting specific cell types or data types in the training data, but maybe this is addressed in the CHM13 sample, which is not included in the training dataset?

Thanks for the above concern and suggestion. As suggested, by using the HG002 CCS datasets (71.0×), we perform an analysis to compare the methylation frequencies calculated by the count mode and model mode of csmeth. We added this analysis in detail in Supplementary Note 2 and Supplementary Fig. 7:

Suppose R_b , R_c , R_m are the methylation frequencies of a CpG calculated by BS-seq, count mode of ccsmeth, and model mode of ccsmeth, respectively. We use $|R_b - R_c| - |R_b - R_m|$ to measure whether R_c or R_m is closer to R_b . If $|R_b - R_c| - |R_b - R_m| > 0.1$, meaning the model mode has a more accurate prediction than count mode, we classify the CpG into the group G_m . If $|R_b - R_c| - |R_b - R_m| < -0.1$, we classify the CpG into the group G_c . We find that among the total tested 29,174,320 CpGs, 3,975,014 CpGs are classified to G_m , while 644,370 CpGs are classified to G_c (Supplementary Fig. 7a). The methylation frequencies of CpGs in the two groups show significant differences: CpGs in G_m tend to have either very high (>0.8) or low (<0.2) methylation frequencies, while CpGs in G_c tend to have intermediate methylation frequencies (Supplementary Fig. 7b).

ccsmeth gets higher performance than primrose in the CHM13 and SD0651_P1 samples, which are not included in the training data. The results on CHM13 and SD0651_P1 conclude that no overfitting is observed in ccsmeth.

For count mode, why do you include very ambiguous calls (those with probability close to 0.5)? I think you would have better accuracy if you filtered out ambiguous reads (for instance, reads with probability 0.33-0.66, as the Oxford Nanopore modbam2bed tools does) <https://github.com/epi2me-labs/modbam2bed> . I would like to see a new version of Fig 3C-D which uses such a filtering, just to see if it improves the accuracy.

Thanks for the suggestion! As suggested, we perform analyses to explore if filtering out the ambiguous calls can improve the accuracy of ccsmeth at both the read level and genome-wide site level.

We define $\Delta_p = |P_r - P'_r|$ to filter out the ambiguous calls, where P_r is the methylation probability, and P'_r is the unmethylated probability defined as $1 - P_r$. We set Δ_p to 0.33 by default as modbam2bed does. The results show that when Δ_p is set to 0.33, the read-level accuracies of ccsmeth improve by 3.5-4.2%, with 8.9-12.8% of calls being discarded.

At the genome-wide site level, when Δ_p is set to 0.33, the Pearson correlations of ccsmeth with BS-seq and nanopore sequencing improve by ~1-2% compared to $\Delta_p = 0$ on the HG002, SD0651_P1, and CHM13 datasets.

We have added the read-level analysis in the *Results* section and Supplementary Fig. 4. We included the site-level analysis in the *Results* section and updated Fig. 3a-d and Supplementary Fig. 5. We also cited modbam2bed in the revised manuscript. We also added an option “--prob_cp” in ccsmeth for users to set a custom Δ_p .

Data availability section states, “The sequencing data of NA12898, SD0651_P1 and the HN0641 family trio will soon be available at NCBI.” It is important that the reviewers can see these datasets (even if they are available). It’s also important to have the methods described (how patient samples were collected and processed, etc.)

Thanks for the concerns. This study is compliant with the “Guidance of the Ministry of Science and Technology (MOST) of China for the Review and Approval of Human Genetic Resources”. We have deposited all sequencing data generated in this study to the Genome Sequence Archive in the National Genomics Data Center, Beijing Institute of Genomics (BIG, <http://gsa.big.ac.cn>), Chinese Academy of Sciences, under Project accession No. PRJCA015556[<https://ngdc.cncb.ac.cn/bioproject/browse/PRJCA015556>]. The sequencing data of NA12898 (GSA-Human accession No. HRA004180[<https://ngdc.cncb.ac.cn/gsa-human/browse/HRA004180>]) is available under open access. The sequencing data of SD0651_P1 and the HN0641 family trio (GSA-human accession No. HRA004202[<https://ngdc.cncb.ac.cn/gsa-human/browse/HRA004202>]) is available under restricted access, which can be granted by the Data Access Committee (DAC). We also sequenced a Zebrafish sample using PacBio CCS and BS-seq. The sequencing data

of Zebrafish has also been deposited to GSA/NGDC under accession No. CRA010412[<https://ngdc.cncb.ac.cn/gsa/browse/CRA010412>].

We added more description about samples collection and genome sequencing in the *Methods* section of the revised manuscript as follows:

“We selected the Chinese sample SD0651_P1 and the Chinese family trio (HN0641_FA, HN0641_MO, HN0641_S1) from the Chinese autism spectrum disorder cohort [1] with no specific sex or age requirements. All the participants signed the informed consent before sample collection. The sequencing of the four samples was approved by the Research Ethics Committee in the School of Life Sciences, Central South University (No. 2021-1-6). The genomic DNA for PacBio sequencing and bisulfite sequencing was extracted from the peripheral blood.”

References

[1] Tianyun Wang, Hui Guo, Bo Xiong, Holly AF Stessman, Huidan Wu, Bradley P. Coe, Tychele N. Turner et al. "De novo genic mutations among a Chinese autism spectrum disorder cohort." *Nature communications* 7, no. 1 (2016): 13316.

Minor

For Fig 3A-D, could you use the same Y axis limits for all plots? That would make it easier to compare.

We have changed the Y-axis in Fig. 3a-d as suggested.

In Fig 5B and 5D, it would be good to show a Venn Diagram for the non-repeatmasker (unique) CpGs as well.

In the revised manuscript, we showed the Venn Diagram for non-repeatmasker CpGs in Supplementary Fig. 19. In non-RepeatMasker regions, PacBio CCS detects methylation states of 96.1% CpGs, which are 8.9% more than the CpGs detected by BS-seq. PacBio CCS also phases 64.4% more CpGs than BS-seq.

p.9: Grammatical fix. “are got from Baid et al” should be “were taken from Baid et al”

We have corrected this accordingly.

p.9: You start by saying that these experiments used NA12898. But then several times you reference NA12878. Was this a typo, or are you actually using both of these two samples? It was not clear.

This was a typo. We have corrected it.

Response to Reviewer #3

Reviewer #3 (Remarks to the Author):

Ni et al. present the work of csmeth for identifying CpG methylation using PacBio CCS data and csmethphase for phasing methylation and identifying differentially methylated regions between haplotypes. Overall, the methods are of great interest to the community, the data appear to be high quality, and the results are well presented. Specifically, they show that csmeth performs better than primrose, another CCS methylation calling program developed by PacBio, and is comparable to nanopore data called by deepsignal2. They also applied the method across different human samples and showed good correlations with the traditional BS-seq data. These are very exciting results for human studies.

We thank the reviewer for the positive comments.

One major concern is the generalizability of the research and the potential for overtraining. For the potential readership of Nature Communications, a large number of researchers are studying non-human non-model species that lack proper training data as their research targets, such as fungi, plants, and non-human vertebrates. It is important to document to what extent their model can be applied to other species, especially since nanopore approaches and even primrose have been shown to be robust with non-human data.

Thank you for the above concern and suggestion. To test csmeth using non-human data, we performed PacBio CCS and BS-seq of a Zebrafish sample in parallel. The genomic DNA was extracted from the muscular tissue of the Zebrafish. We got $23.3\times$ CCS reads and $29.5\times$ BS-seq reads in total. We used the pre-trained model of csmeth to detect 5mCpG methylation from the CCS reads of the Zebrafish sample and compare the results with BS-seq. We also detected 5mCpG methylation from the CCS reads using primrose for comparison. The results show that csmeth gets higher correlations with BS-seq than primrose (0.8463 vs. 0.8292), which demonstrate the robustness of csmeth for detecting methylation from non-human data. The relatively low correlation may be due to the low coverage of BS-seq data.

We have added this analysis in Supplementary Fig. 20 and the Discussion section of the revised manuscript.

Minor:

1. Page 6, paragraph 2, the authors claimed csmethphase could be the first haplotype-aware methylation calling tool. This may not be true. For example, modbamtools (<https://rrazaghi.github.io/modbamtools/>) developed by Razaghi et al. (2022), can identify and visualize CpG haplotypes.

Thanks for pointing this out. We have deleted the corresponding sentence and cited modbamtools in the revised manuscript.

2. End of page 7 - beginning of P8, “Then the feature matrix is fed into one BiGRU layer and then into one attention layer” could benefit from more context and details.

We rewrote this sentence to “The feature matrix is first input into a BiGRU layer to capture the forward and reverse flow of interactions between adjacent CpGs. Then an attention layer is applied to optimize the weights of each adjacent CpG, which allows the model to focus on the most relevant interactions”.

3. Figure 1a, the orange bars are not labeled.

Thanks for pointing this out. The orange bars are adapters. We have labeled them in the revised manuscript.

4. Figure 3f, the correlation between ccsmeth and BS-seq in the SD0651_P1 sample is relatively low ($r = 0.8750$) compared to HG002 (Figure 3e, $r \geq 0.9271$), can you comment on this?

In the BS-seq data for SD0651_P1, there is only 15.7× coverage of reads, while the BS-seq data for HG002 has 117.5× coverage of reads. To investigate whether the coverage of BS-seq reads affects the correlation between BS-seq and ccsmeth, we subsampled 15.7× reads from the total BS-seq data of HG002 for comparison. As shown in Table R1, the correlation between BS-seq and ccsmeth significantly decreases as the coverage of BS-seq reads decreases.

Table R1. Correlations between methylation frequencies predicted by ccsmeth and BS-seq for HG002 under difference coverages.

BS-seq read coverage	CCS reads coverage of HG002 15Kb					
	5×	10×	15×	20×	25×	25.6×
117.5×	0.8993	0.9198	0.9326	0.9403	0.9453	0.9463
15.7×	0.8679	0.8890	0.9029	0.9111	0.9164	0.9174

5. Suppl. Fig. 9a, can you comment on why ccsmeth identified DMRs are more distant away from the BS-seq identified DMRs?

Thanks for the concern. We believe this is related to the inability of BS-seq to phase the majority of CpGs in the genome. The short read limits the ability of BS-seq for methylation phasing. As shown in Fig. 5c, only 6.71M CpGs (coverage ≥ 5) are phased to two haplotypes by BS-seq. Based on the phased CpGs, only 2,463 DMRs are identified. However, when using CCS and nanopore data, 14,390 and 16,250 DMRs are identified, respectively. Therefore, the DMRs identified by ccsmeth are more distant away from those identified by BS-seq.

6. For tests involving multiple replications, plotting the standard deviation should be helpful to contextualize the variation in program performance. For example, Fig2, Fig3a-d, Fig 5ac, and several suppl Figs.

Thanks for the suggestion. In the revised manuscript, we have added the standard deviation information of tests involving multiple replications. In Fig. 2, Fig. 3a-d, Fig. 5ac, and Supplementary Fig. 5, the standard deviations are too small to be visible in the plots. Therefore, we included the standard deviations in Supplementary Tables 4-12, 14-15, and Supplementary Data 1. We have also clarified this in the caption of each figure.

Reviewer #2 (Remarks to the Author):

The authors performed significant new analyses and put in diligent work to address all my concerns. I am satisfied with their revised manuscript and they addressed all concerns sufficiently. While I am satisfied with this revision, the new results in Supplementary Fig. 7 are still hard to interpret. I would suggest that they add a (C) panel which is the scatter plot or scatter cloud showing the methylation values of the count mode on one axis and the methylation values of the model mode on the other axis. I remain concerned that the model mode may be overfitting the training data close to 0 and 1 in ways that are not desirable, and could perform poorly in new cell types or genomic elements with intermediate methylation. So I would still urge caution using this mode until it has been tested under more conditions. But I do think at this point that you have provided enough validation data to support cautious use of this model.

Ben Berman
Hebrew University of Jerusalem

Reviewer #3 (Remarks to the Author):

The revised manuscript has addressed all my comments. Thank you for developing ccsmeth and ccsmethphase!

The response to Reviewer #1's comments was assessed by Reviewer #3:

I carefully read the author's response to reviewer 1's comments, and found they addressed them very well.

Summary

We appreciate the valuable comments and suggestions from the editor and reviewers. Based on the suggestions and comments from editor and reviewers, we revised our paper. We addressed those comments and suggestions carefully. The significant changes in the revised manuscript were highlighted in red color.

Response to Reviewer #2

Reviewer #2 (Remarks to the Author):

The authors performed significant new analyses and put in diligent work to address all my concerns. I am satisfied with their revised manuscript and they addressed all concerns sufficiently.

While I am satisfied with this revision, the new results in Supplementary Fig. 7 are still hard to interpret. I would suggest that they add a (C) panel which is the scatter plot or scatter cloud showing the methylation values of the count mode on one axis and the methylation values of the model mode on the other axis. I remain concerned that the model mode may be overfitting the training data close to 0 and 1 in ways that are not desirable, and could perform poorly in new cell types or genomic elements with intermediate methylation. So I would still urge caution using this mode until it has been tested under more conditions. But I do think at this point that you have provided enough validation data to support cautious use of this model.

Authors' Response. We thank the reviewer for the supportive and helpful comments! As suggested, we have added a (C) panel in Supplementary Fig. 7 (See also Fig. R1 below) to compare the methylation values of the count mode and model mode of csmeth. In future research, we will continue to test csmeth in more cell types.

Fig. R1 Comparison of genome-wide per-site methylation frequency between the count mode and model mode of csmeth.